# LEO-VL: Efficient Scene Representation for Scalable 3D Vision-Language Learning

Figure 1: **An overview of LEO-VL.** LEO-VL features an efficient scene representation with strong perception capability and low representation costs. This unlocks the scalability of 3D vision-language (3D-VL) learning across diverse scene domains (*e.g.*, ScanNet, 3RScan) and tasks such as captioning, dialogue, *etc*.

## ABSTRACT

Developing vision-language models (VLMs) capable of understanding 3D scenes has been a longstanding goal in the 3D-VL community. Despite recent progress, 3D VLMs still fall short of their 2D counterparts in capability and robustness. A key bottleneck is that current scene representations struggle to balance performance and efficiency: competitive performance comes at the cost of heavy token overhead, which in turn hampers the scalability of 3D-VL learning. To address this, we propose the condensed feature grid (CFG), an efficient scene representation featuring significantly reduced token overhead and strong perception capability. Building on CFG, we introduce LEO-VL, a 3D VLM trained on 700k 3D-VL data spanning four real-world indoor domains and five tasks such as captioning and dialogue. To enhance the robustness of 3D VLM, we further propose SceneDPO for post-training, which involves contrasts across answers and scenes. LEO-VL achieves state-of-the-art performance on various 3D QA benchmarks, including SQA3D, MSQA, and Beacon3D. Our extensive experiments highlight the efficiency of our representation, the benefit of task and scene diversity, consistent scaling effects, and the advantages of SceneDPO compared to SFT and GRPO. We hope our findings advance the efficiency, scalability, and robustness of future 3D VLMs.

## 1 INTRODUCTION

With the crucial role of 3D scene understanding in human intelligence, building 3D VLMs that can understand 3D environments and communicate with humans through natural language has become a central goal of the 3D-VL community (Huang et al., 2024b; Yang et al., 2025b; Song et al., 2025; Ma et al., 2024). Recent progress has advanced capabilities across 3D-VL tasks such as grounding (Zhu et al., 2023; 2024c; Jia et al., 2024), captioning (Luo et al., 2023; Xu et al., 2024), reasoning (Huang et al., 2024a; Linghu et al., 2024), and dialogue (Huang et al., 2024b; Yang et al., 2024). Nonetheless, current 3D VLMs fall significantly short of the capabilities exhibited by their 2D counterparts (OpenAI, 2023; Team et al., 2023; Wang et al., 2024b; Chen et al., 2024c; Li et al., 2024a). We argue that a key obstacle to advancing 3D VLMs lies in scene representation, which struggles in the performance-efficiency trade-off and could hinder the scalability of 3D-VL learning.

Existing scene representations for 3D VLMs can be categorized into two approaches. The first adopts 3D modality input such as point cloud, requiring complex pre-processing pipelines such as 3D reconstruction and instance segmentation (Zhu et al., 2023; Jia et al., 2024). This approach also confronts the inherent difficulty of 3D perception, which is challenging given the scarcity of 3D-VL data. The second approach uses 2D images or videos as input, leveraging the strength of 2D perception (Zhu et al., 2024b; Zheng et al., 2025b). Despite stronger performance, it incurs substantial token overhead and severely limits data scalability. Additionally, both approaches typically adopt rudimentary 3D position embeddings, which may fall short in modeling complex spatial structures.

To overcome these limitations, we propose the condensed feature grid (CFG), a novel scene representation that significantly improves efficiency while preserving strong perception capability. Given multi-view RGB-D inputs, we back-project 2D features to 3D voxels, encode their height features using rotary position embedding (RoPE) (Su et al., 2024), and vertically condense voxels within each pillar into grid tokens. We further inject horizontal position embeddings with 2D Fourier features (Tancik et al., 2020). This design reduces scene token overhead to only 33%, while maintaining accurate 3D structures. Based on this representation, we build our model LEO-VL by integrating the CFG tokens with a large language model (LLM) for auto-regressive language modeling.

As the representation efficiency unlocks scalability of 3D-VL learning, we construct a comprehensive 3D-VL dataset spanning four real-world indoor domains (Dai et al., 2017; Wald et al., 2019; Mao et al., 2022; Baruch et al., 2021) and five 3D-VL tasks. We curate existing data and generate new samples where necessary. We emphasize data quality over scale to avoid model performance degradation, and ultimately collect over 700k diverse and high-quality data samples for 3D-VL instruction tuning.

Notably, prior studies suggest that supervised finetuning (SFT) alone can be insufficient to build robust 3D VLMs given the prevalent overfitting risks (Deng et al., 2024; Huang et al., 2025b). To enhance the robustness of 3D VLMs, we propose SceneDPO, a post-training objective that involves contrasts across answers and scenes. We further demonstrate that SceneDPO yields better in-domain and out-of-domain performances compared to SFT and group relative policy optimization (GRPO).

We show that LEO-VL achieves state-of-the-art performance with significantly higher efficiency on various 3D QA benchmarks, including ScanQA, SQA3D, MSQA, and Beacon3D. Ablation studies highlight the efficiency and spatial modeling of CFG, the impact of task diversity such as captioning and dialogue, and the importance of scene diversity for strong cross-domain performance. Moreover, we observe that naive scaling with low-quality QA data degrades performance, whereas our data curation principle yields consistent scaling effects.

In summary, our contributions are as follows:

1. We propose LEO-VL, a 3D VLM built on condensed feature grid (CFG), an efficient scene representation featuring strong perception capability and simplified 3D spatial modeling.
2. We construct a comprehensive 3D-VL instruction-tuning dataset of 700k samples across four domains and five tasks, prioritizing data quality to ensure reliable scaling for 3D-VL learning.
3. We demonstrate LEO-VL's state-of-the-art performance across various 3D QA benchmarks, supported by extensive ablations on representation efficiency, task and domain diversity, and data quality. We further introduce SceneDPO as an effective post-training objective for 3D VLM.

## 2 RELATED WORK

**3D Vision-Language Models.** Most early works in 3D-VL understanding develop models based on 3D representations such as point clouds (Qi et al., 2017a;b; Phan et al., 2018; Li et al., 2018; Wu et al., 2019; Qi et al., 2019; Zhao et al., 2021a; Huang et al., 2021; Yu et al., 2022) and voxels (Graham, 2015; Maturana & Scherer, 2015; Riegler et al., 2017; Tatarchenko et al., 2017; Graham et al., 2018; Dai & Nießner, 2018; Choy et al., 2019; Schult et al., 2023). With the maturation of 3D-VL pretraining and instruction tuning techniques, 3D VLMs have demonstrated significant improvements in capabilities (Zhao et al., 2021b; Huang et al., 2022; Abdelreheem et al., 2022; Chen et al., 2022; Huang et al., 2022; Liu et al., 2023b; Xue et al., 2024; Zhu et al., 2025; 2023; Zhou et al., 2024; Wang et al., 2025; Hong et al., 2023; Xu et al., 2024; Huang et al., 2024b; Yang et al., 2024; Chen et al., 2024a; Qi et al., 2024; Zhu et al., 2024c; Fu et al., 2025; Zhang et al., 2024b; Chu et al., 2024; Huang et al., 2024a; Deng et al., 2025). In contrast, an alternative line of work leverages 2D perception models to handle 3D-VL tasks and exhibits strong performances (Peng et al., 2023; Shafiullah et al., 2023; Ding et al., 2023; Jatavallabhula et al., 2023; Huang et al., 2023; El Banani

et al., 2024; Man et al., 2024; Luo et al., 2025; Zhu et al., 2024b; Zheng et al., 2025b; Qi et al., 2025). However, adapting 2D VLMs to 3D scene understanding poses two challenges: the cost of representing 3D scenes and the absence of effective 3D spatial modeling. Despite recent efforts (Zhi et al., 2025; Yu et al., 2025; Thai et al., 2025; Zheng et al., 2025a; Huang et al., 2025c), the scene representation still struggles with a performance-efficiency trade-off (Huang et al., 2025a; Zhang et al., 2025b). In contrast, we propose a novel scene representation equipped with disentangled 3D spatial modeling, which preserves both strong perception capability and high efficiency, delivering an important direction and practical solution for improving the efficiency of 3D VLMs.

**3D Vision-Language Datasets and Benchmarks.** Based on indoor 3D scene assets (Dai et al., 2017; Yeshwanth et al., 2023; Wald et al., 2019; Mao et al., 2022; Baruch et al., 2021; Chang et al., 2017; Ramakrishnan et al., 2021; Chang et al., 2015; Khanna et al., 2024; Zheng et al., 2020; Fu et al., 2021; Deitke et al., 2022), existing 3D-VL datasets primarily focus on object grounding (Chen et al., 2020; Achlioptas et al., 2020; Zhang et al., 2023) and question answering (QA) (Azuma et al., 2022; Ma et al., 2023) tasks. As the 3D-VL community advances, recent efforts have focused on aggregating diverse scene domains to enable large-scale pretraining (Zhu et al., 2023; 2024c; Jia et al., 2024; Wang et al., 2024c) and instruction tuning (Huang et al., 2024b), followed by more cross-domain datasets (Yang et al., 2025a; Lyu et al., 2024; Linghu et al., 2024; Song et al., 2025; Zhang et al., 2025a) and benchmarks (Linghu et al., 2024; Huang et al., 2025b; Yang et al., 2025b). Given the potential of cross-domain scaling for 3D-VL learning, we compile a comprehensive 3D-VL data scheme covering four real-world scene domains and five instruction-tuning tasks. We further demonstrate the benefits of domain and task diversity through comprehensive evaluations.

**Post-training for Vision-Language Models.** Prior studies (Deng et al., 2024; Huang et al., 2025b; Peng et al., 2025; Zhang et al., 2025c) suggest that SFT, as a common strategy for 3D VLMs, may undergo weak robustness given the scarce data and overfitting risk in 3D-VL learning. To address this, existing efforts mainly involve data augmentation (Yang et al., 2025a; Kang et al., 2024; Zhao et al., 2024), with the learning objective underexplored. Motivated by the success of reinforcement learning from human feedback (RLHF) (Ouyang et al., 2022; Schulman et al., 2017; Rafailov et al., 2023; Yuan et al., 2024; Pang et al., 2024; Liu et al., 2024; Chu et al., 2025), recent works (Li et al., 2024b; Pi et al., 2024; Wang et al., 2024a; Xie et al., 2024) demonstrate the efficacy of direct preference optimization (DPO) (Rafailov et al., 2023) on 2D VLMs. In contrast, we introduce SceneDPO as a novel and effective post-training objective tailored for 3D VLM.

## 3 METHOD

In Section 3.1, we introduce the model design of LEO-VL, which features an efficient representation of 3D scenes. In Section 3.2, we introduce the recipe of our comprehensive 3D-VL instruction-tuning datasets, which spans four real-world indoor domains and five tasks. In Section 3.3, we further introduce a novel post-training objective to enhance the robustness of 3D VLMs.

### 3.1 MODEL

**Overview.** As illustrated in Fig. 2, given multi-view RGB-D images, LEO-VL extracts 2D visual features and transforms them into the condensed feature grid (CFG). Next, an LLM performs auto-regressive language modeling over the joint sequence of CFG tokens and language tokens. In contrast to prior object-centric 3D VLMs (Huang et al., 2024b; Zhu et al., 2024c; Huang et al., 2024a; Linghu et al., 2024), LEO-VL employs 2D perception to address limitations such as the dependency of object masks, scene-level information loss, and poor perception capability. In contrast to recent video-based VLMs (Zhu et al., 2024b; Zheng et al., 2025b; Qi et al., 2025), LEO-VL substantially reduces the representation overhead and simplifies 3D spatial modeling.

**2D Perception and Back-projection.** We employ a 2D visual encoder to extract image features from multi-view RGB inputs. Given a set of RGB images $\{I_i \in \mathbb{R}^{H \times W \times 3}\}_{i=1}^{N}$, the encoder produces feature maps $\{M_i \in \mathbb{R}^{h \times w \times d}\}_{i=1}^{N}$, where $h$, $w$, and $d$ denote the resolution and feature dimension. The corresponding depth maps $\{D_i \in \mathbb{R}^{H \times W}\}_{i=1}^{N}$ are interpolated to patch resolution $h \times w$ and then back-projected into 3D space. These 3D points will be assigned 2D features $M_i$. Let $K \in \mathbb{R}^{3 \times 3}$ and $T \in \mathbb{R}^{4 \times 4}$ denote the intrinsic and extrinsic camera matrices, respectively. For example, a pixel at coordinate $q = (i, j)$ with depth $D(q)$ is back-projected to a 3D point $(x, y, z)$ as follows:

$$\bar{p} = T \left[ [D(q)K^{-1}\bar{q}]^{\mathrm{T}} \parallel 1 \right]^{\mathrm{T}}, \text{ where } \bar{p} = [x, y, z, 1]^{\mathrm{T}} \text{ and } \bar{q} = [i, j, 1]^{\mathrm{T}}.$$

Figure 2: **Overview of LEO-VL model.** LEO-VL extracts 2D visual features from multi-view RGB-D frames and transforms the features into CFG, significantly reducing the token overhead while preserving 3D spatial structure. An LLM performs auto-regressive language modeling based on the CFG tokens and language tokens.

**Voxelization.** We first voxelize the 3D points by averaging point features within each voxel, with empty voxels discarded. For referring tasks such as object captioning, a learnable anchor embedding is added to voxels that fall within the referred region. This yields an initial voxel-based representation $v_{(x,y,z)}$, which remains computationally expensive and poses challenges for 3D spatial modeling.

**Condensed Feature Grid.** To further improve efficiency, our core is to condense voxels into a 2D planar grid by pooling voxel features vertically within each pillar. To address the challenge of 3D spatial modeling, we disentangle the position encoding along vertical and horizontal directions, respectively. The process of converting voxels into CFG consists of three stages.

- **Vertical Position Encoding.** We encode the vertical position (*i.e.*, height) of each voxel using RoPE (Su et al., 2024), which applies a rotation-like transformation to the feature. The rotation angles relate to a pre-defined channel-wise frequencies and position value (*i.e.*, height). Compared to additive position embeddings (Vaswani et al., 2017; Tancik et al., 2020), the rotary nature enables RoPE to better capture vertical spatial relations, such as distinguishing objects at different heights (see example in *Appendix*). The height-encoded voxel feature $\bar{v}_{(x,y,z)}$ is formulated as:

$$\bar{v}_{(x,y,z)} = R(z)v_{(x,y,z)}.$$

- **Vertical Condensation.** Let $C(x^*, y^*)$ denote the count of voxels located in the pillar at $(x^*, y^*)$. For each horizontal position $(x^*, y^*)$ where $C(x^*, y^*) > 0$, the conversion from voxels $\bar{v}_{(x,y,z)}$ to CFG token $g_{(x^*,y^*)}$ is formulated as:

$$g_{(x^*,y^*)} = \frac{1}{C(x^*, y^*)} \sum_{(x,y,z)} \bar{v}_{(x,y,z)} \mathbb{1}(x = x^*, y = y^*).$$

- **Horizontal Position Encoding.** We encode the horizontal position $(x^*, y^*)$ with 2D Fourier features (Tancik et al., 2020), which applies a linear transformation (denoted by $W$) to the position value $(x^*, y^*)$ and then computes sinusoidal components, followed by an MLP to enhance expressiveness. Let $\mathbf{p} = (x^*, y^*)$ denote the horizontal position, the final CFG token $\bar{g}_{(x^*,y^*)}$ is formulated as:

$$\bar{g}_{(x^*,y^*)} = g_{(x^*,y^*)} + \text{MLP}\big( [\cos(2\pi W\mathbf{p}) \| \sin(2\pi W\mathbf{p})] \big).$$

**Large Language Model.** An LLM auto-regressively generates text output based on the joint sequence of CFG tokens $\bar{g}_{(x^*,y^*)}$ and instruction tokens. The CFG scene tokens are directly used without any resampling process (Zhu et al., 2024c; Chen et al., 2024a) owing to low token overhead.

### 3.2 DATA

The construction of LEO-VL training data is guided by three principles: (1) the diversity of scene domains, which demonstrates crucial for 3D-VL learning (Jia et al., 2024); (2) the diversity of tasks, which we think necessary given the benefits of long-response tasks (Huang et al., 2024b; Lyu et al., 2024); and (3) the balance between quality and scale, considering the potential risks in hindering model performance from low-quality data. We detail the composition of our data in Table 1.

**Scene Domains.** We include four real-world indoor domains: ScanNet (Dai et al., 2017), 3RScan (Wald et al., 2019), MultiScan (Mao et al., 2022), and ARKitScenes (Baruch et al., 2021), following prior practices in scaling 3D scenes (Jia et al., 2024; Linghu et al., 2024). We exclude scene domains that lack attribute-rich scene graphs (*e.g.*, HM3D (Ramakrishnan et al., 2021)), which are essential for LLM-assisted data generation.

Table 1: **Overview of LEO-VL training data**. The entries include: "SV" for SceneVerse (Jia et al., 2024), "MM" for MMScan (Lyu et al., 2024), ScanQA (Azuma et al., 2022), "SQA" for SQA3D (Ma et al., 2023), MSQA (Linghu et al., 2024), LEO (Huang et al., 2024b), and "✓" for newly created data in this work.

| | ScanNet | 3RScan | MultiScan | ARKitScenes | Overall count | Avg. length (str) | Avg. length (words) |
|---|---|---|---|---|---|---|---|
| ObjCap | SV, MM | LEO, SV, MM | SV | SV | 216k | 299 | 56 |
| SceneCap | MM, ✓ | LEO, MM, ✓ | ✓ | ✓ | 128k | 633 | 104 |
| QA | ScanQA, SQA, MSQA | MSQA | - | MSQA | 289k | 31 | 6 |
| Plan | ✓ | LEO | ✓ | ✓ | 18k | 534 | 96 |
| Dialog | ✓ | LEO | ✓ | ✓ | 61k | 93 | 18 |

**Tasks and Datasets.** We incorporate five prevalent 3D-VL tasks with natural language outputs, which are compatible with a unified instruction tuning framework. We exclude the 3D object grounding task due to its different formulation. We think the absence of the grounding task is not detrimental, given the success of 2D VLMs without explicit grounding supervision (Liu et al., 2023a; OpenAI, 2023; Wang et al., 2024b; Chen et al., 2024c). In addition to existing datasets, we generate new data by prompting LLMs with scene graphs (Huang et al., 2024b; Linghu et al., 2024) to fill the blank ("✓" entries) in Table 1. Our data covers five categories of text-output tasks:

- **Object Captioning.** This task is to describe a specific object in natural language. We adopt object captions from SceneVerse (Jia et al., 2024) and MMScan (Lyu et al., 2024). We exclude datasets that re-purpose object grounding texts as captions (Chen et al., 2021; Achlioptas et al., 2020), as they lack diverse descriptions regarding object attributes.
- **Scene Captioning.** This task requires generating comprehensive descriptions of 3D scenes. We use scene captions from LEO (Huang et al., 2024b) and MMScan (Lyu et al., 2024), and generate situated scene captions that incorporate situations to resolve spatial ambiguities (*e.g.*, left and right).
- **Question Answering.** This task requires answering general questions about the scene. We include ScanQA (Azuma et al., 2022), SQA3D (Ma et al., 2023), and MSQA (Linghu et al., 2024). We do not generate extra QA data as we find it can exhibit trivial patterns and degrade model performance.
- **Planning.** This task is to generate a step-by-step grounded plan for a high-level goal (*e.g.*, "organize the room"). We use the planning data from LEO (Huang et al., 2024b) for 3RScan, and generate new data for other scene domains.
- **Dialogue.** This task concerns generating responses conditioned on both the 3D scene and dialogue context. We use the dialogue data from LEO (Huang et al., 2024b) for 3RScan, and generate new data for other scene domains.

## 3.3 POST-TRAINING

We argue that only SFT can be insufficient to build robust 3D VLMs, as evidenced by prior studies (Li et al., 2024b; Pi et al., 2024; Wang et al., 2024a). In particular, given the pronounced risk of overfitting in 3D VLMs (Deng et al., 2024; Huang et al., 2025b), it is crucial to design an effective post-training objective to enhance their robustness.

To this end, we propose SceneDPO, a novel post-training objective for 3D VLMs. We start with a DPO-like loss term $\mathcal{L}_a$ that contrasts between positive answer $a_{\checkmark}$ and negative answer $a_{\chi}$. Motivated by the issue of visual ignorance (Huang et al., 2025b), we introduce a loss term $\mathcal{L}_s$ that contrasts between positive scene $s_{\checkmark}$ and negative scene $s_{\chi}$. This discourages the model from predicting the current answer when conditioned on irrelevant scenes. To mitigate degradation of positive answers, we incorporate a negative log-likelihood loss term $\mathcal{L}_{\mathrm{NLL}}$, which proves critical in prior works (Liu et al., 2024; Pang et al., 2024; Wang et al., 2024a). Given a training tuple $(s_{\checkmark}, s_{\chi}, q, a_{\checkmark}, a_{\chi}) \sim \mathcal{D}$, the overall loss $\mathcal{L}$ is defined as follows:

$$\mathcal{L}_a = \mathbb{E}_{\mathcal{D}} \left[ -\log \sigma \left( \beta_a \log \frac{\pi_\theta(a_{\checkmark} \mid s_{\checkmark}, q)}{\pi_{\mathrm{ref}}(a_{\checkmark} \mid s_{\checkmark}, q)} - \beta_a \log \frac{\pi_\theta(a_{\chi} \mid s_{\checkmark}, q)}{\pi_{\mathrm{ref}}(a_{\chi} \mid s_{\checkmark}, q)} \right) \right], \mathcal{L}_{\mathrm{NLL}} = \mathbb{E}_{\mathcal{D}} \left[ -\log \pi_\theta(a_{\checkmark} \mid s_{\checkmark}, q) \right],$$

$$\mathcal{L}_s = \mathbb{E}_{\mathcal{D}} \left[ -\log \sigma \left( \beta_s \log \frac{\pi_\theta(a_{\checkmark} \mid s_{\checkmark}, q)}{\pi_{\mathrm{ref}}(a_{\checkmark} \mid s_{\checkmark}, q)} - \beta_s \log \frac{\pi_\theta(a_{\checkmark} \mid s_{\chi}, q)}{\pi_{\mathrm{ref}}(a_{\checkmark} \mid s_{\chi}, q)} \right) \right], \quad \mathcal{L} = w_a \mathcal{L}_a + w_s \mathcal{L}_s + \mathcal{L}_{\mathrm{NLL}},$$

where $\theta$ denotes the trained model, ref denotes the reference model, $\sigma$ denotes the sigmoid function, and $w_a$, $w_s$, $\beta_a$, and $\beta_s$ are scalar hyperparameters.

# 4 EXPERIMENT

We first compare LEO-VL against state-of-the-art models on various 3D QA benchmarks in Section 4.1, highlighting its performance and efficiency. In Section 4.2, we present ablation studies on model design to show the effectiveness of CFG. In Section 4.3, we analyze various training data configurations to reveal the influence of task diversity, scene domain coverage, data quality, and data scale on 3D-VL learning. In Section 4.4, we further explore the post-training for 3D VLM.

**Implementation Details.**   We initialize the 2D visual encoder and LLM with Qwen2.5-VL-7B-Instruct (Bai et al., 2025). The learnable parameters include position embedding, anchor embedding, and LoRA parameters (Hu et al., 2022) of the LLM, which amount to 66M in total. We set the voxel size to 0.2 meters and retain up to 750 scene tokens for CFG. We train LEO-VL on our comprehensive 3D-VL dataset for 5 epochs, which takes 2 days with 8 NVIDIA A100 80G GPUs. We adopt AdamW optimizer (Loshchilov & Hutter, 2017) with a base learning rate at $3 \times 10^{-5}$, scheduled with linear warmup and cosine decay. We provide detailed hyperparameters in *Appendix*.

**Evaluation.**   Our evaluation mainly involves 3D QA, including ScanQA (Azuma et al., 2022) for general QA, SQA3D (Ma et al., 2023) for situated QA, MSQA (Linghu et al., 2024) for situated QA across multiple scene domains, and Beacon3D (Huang et al., 2025b) for robust QA evaluation in diverse scene domains. For the captioning task, we report the evaluation on Scan2Cap (Chen et al., 2021) in *Appendix*, as we think existing captioning benchmarks fall short in reflecting natural descriptive capabilities. Evaluation of other tasks is not reported due to the limited availability of standardized benchmarks. We report n-gram metrics (CIDEr and BLEU-4) for ScanQA, exact-match accuracy (EM and EM-R) for SQA3D, and GPT-Score for MSQA and Beacon3D.

## 4.1 COMPARISON WITH STATE-OF-THE-ART MODELS

**Baselines.**   For comparison, we include state-of-the-art 3D VLMs across four categories: (1) query-based methods, ranging from early work like ScanQA (Azuma et al., 2022) to recent 3D-LLaVA (Deng et al., 2025); (2) object-centric methods, from 3D-VisTA (Zhu et al., 2023) to Inst3D-LMM (Yu et al., 2025); (3) voxel-based methods, such as Scene-LLM (Fu et al., 2025) and LLaVA-3D (Zhu et al., 2024b); and (4) video-based methods, including Video-3D LLM (Zheng et al., 2025b) and GPT4Scene (Qi et al., 2025). We also show their scene representations and associated costs. For Scene-LLM, only the voxel size (0.18 meters) is available, which we estimate to yield a similar number of tokens as LLaVA-3D (voxel size 0.2 meters). For video-based methods, we list the number of frames, noting that 32 frames typically amount to over 6k scene tokens.

**Results.**   As shown in Tables 2 and 3, LEO-VL achieves state-of-the-art performance on most 3D QA benchmarks with a much lower representation cost. While video-based models show competitive performance, their representations involve thousands of tokens and limit data scalability, hindering scalable 3D-VL learning across more scene domains. In contrast, the efficient representation of LEO-VL enables 3D-VL data scaling for consistently strong performance across domains, including ScanNet, 3RScan, and MultiScan. These results underscore the advantages of our representation in efficiency, perception capability, and data scalability.

## 4.2 MODEL ANALYSIS

**Statistics.**   In Fig. 4, we present statistics of the number of voxel tokens and CFG tokens on ScanNet. In addition to the distribution of token counts, we report two metrics: **compression rate**, defined as the ratio of CFG tokens to raw voxel tokens before condensation; and **preservation rate**, defined as the proportion of voxels retained by the CFG tokens. The results show that CFG achieves an average compression rate of about 33% and an average preservation rate close to 99%. This demonstrates the high efficiency of our CFG representation.

**Efficiency *vs*. Accuracy.**   We provide a joint visualization of efficiency and accuracy in Fig. 3. Accuracy is measured using the EM on SQA3D, the metric with the most available reference results. As shown in Fig. 3, LEO-VL achieves a Pareto optimum between efficiency and accuracy. To investigate the efficiency of CFG, we ablate the representation with two alternatives on the ScanNet subset: **voxel-full**, which retains all voxel tokens without condensation; and **voxel-sample**, which downsamples voxel tokens to match the token count of CFG. The results in Fig. 3 show that compared

Table 2: **Results on 3D QA benchmarks.** Benchmarks are colorized according to scene domains: ScanNet, 3RScan, and MultiScan. "C" stands for "CIDEr", "B-4" for BLEU-4, "EM" for top-1 exact match, "EM-R" for refined top-1 exact match (Huang et al., 2024b), and "Obj." for object-centric metrics (Huang et al., 2025b).

| Model | Scene (#tokens) | ScanQA (val) | | SQA3D (test) | | Beacon3D | | Beacon3D | | Beacon3D | |
|---|---|---|---|---|---|---|---|---|---|---|---|
| | | C | B-4 | EM | EM-R | Case | Obj. | Case | Obj. | Case | Obj. |
| ScanQA (Azuma et al., 2022) | Query (256) | 64.9 | 10.1 | 47.2 | - | - | - | - | - | - | - |
| 3D-LLM (Hong et al., 2023) | Query (32) | 74.5 | 12.9 | 49.8 | - | - | - | - | - | - | - |
| PQ3D (Zhu et al., 2024c) | Query (80) | - | - | 47.1 | - | 35.9 | 4.2 | 25.7 | 0.7 | 20.8 | 0.6 |
| DSPNet (Luo et al., 2025) | Query (256) | - | - | 50.4 | - | - | - | - | - | - | - |
| 3D-LLaVA (Deng et al., 2025) | Query (100) | 92.6 | **17.1** | 54.5 | 56.6 | - | - | - | - | - | - |
| 3D-VisTA (Zhu et al., 2023) | Object (80) | 69.6 | 10.4 | 48.5 | - | 43.2 | 7.3 | 25.7 | **3.3** | 19.1 | 0.0 |
| LEO (Huang et al., 2024b) | Object (60) | **101.4** | 13.2 | 50.0 | 52.4 | 45.2 | 7.5 | 44.0 | 1.5 | 26.2 | 0.6 |
| SceneVerse (Jia et al., 2024) | Object (80) | - | - | 49.9 | - | 40.5 | 4.7 | 37.4 | 0.4 | 28.9 | 3.1 |
| Chat-Scene (Huang et al., 2024a) | Object (200) | 87.7 | 14.3 | 54.6 | 57.5 | 49.8 | 10.9 | - | - | - | - |
| Inst3D-LMM (Yu et al., 2025) | Object (200) | 88.6 | 14.9 | - | - | - | - | - | - | - | - |
| Scene-LLM (Fu et al., 2025) | Voxel (0.18m) | 80.0 | 11.7 | 53.6 | - | - | - | - | - | - | - |
| LLaVA-3D (Zhu et al., 2024b) | Voxel (3096) | 91.7 | 14.5 | 55.6 | 57.6 | 59.1 | 19.0 | - | - | - | - |
| Video-3D LLM (Zheng et al., 2025b) | Video (32 frames) | 100.5 | 16.3 | 57.7 | - | 59.0 | 17.9 | - | - | - | - |
| GPT4Scene (Qi et al., 2025) | Video (32 frames) | 96.3 | 15.5 | 59.4 | 62.4 | 57.2 | 17.9 | - | - | - | - |
| LEO-VL | Grid (750) | 100.4 | 15.5 | **60.8** | **63.7** | **59.5** | **19.2** | **48.2** | **3.3** | **37.7** | **6.9** |

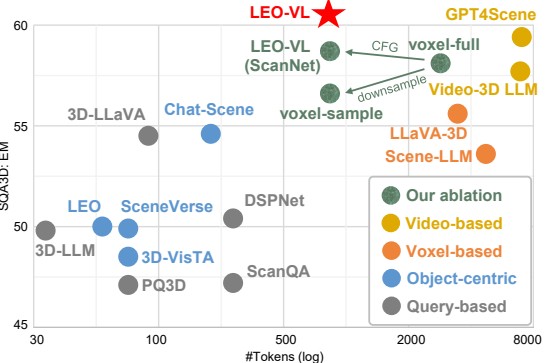

Figure 3: **Efficiency *vs*. accuracy on SQA3D.** LEO-VL reaches a Pareto optimum between efficiency and accuracy.

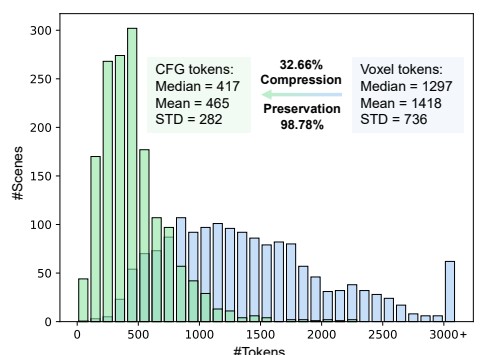

Figure 4: **Statistics of the number of CFG tokens and voxel tokens on ScanNet.**

Table 3: **Detailed results on MSQA (ScanNet) test set.** * indicates text-only input, ‡ indicates zero-shot test, and † indicates results on initial data version.

| Model | Count. | Exist. | Attr. | Spatial | Navi. | Others | Overall |
|---|---|---|---|---|---|---|---|
| GPT-4o*‡ (OpenAI, 2023) | 32.3 | 79.3 | **79.0** | 37.0 | 31.7 | **91.6** | 52.3 |
| LEO (Huang et al., 2024b) | 32.5 | 88.5 | 58.7 | 44.2 | 39.6 | 81.4 | 54.8 |
| MSR3D (Linghu et al., 2024) | 32.3 | **93.1** | 50.0 | 46.5 | 54.1 | 75.6 | 54.2 |
| SplatTalk†‡ (Thai et al., 2025) | 19.6 | 60.3 | 44.0 | 35.8 | 35.5 | 61.8 | 41.8 |
| LEO-VL (ScanNet) | 32.0 | 92.7 | 59.2 | 57.3 | 48.1 | 83.5 | 58.4 |
| LEO-VL (All) | **39.3** | 92.7 | 56.9 | **59.3** | **59.7** | 82.8 | **61.7** |

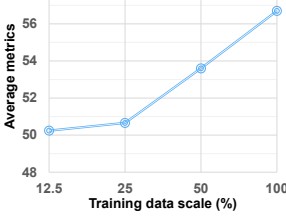

Figure 5: **Scaling curve.**

to voxel-full, voxel-sample suffers a drop in accuracy, while CFG even slightly improves the accuracy. These results suggest that high accuracy in 3D-VL tasks can be achieved with significantly lower token costs, as enabled by our efficient CFG representation.

**3D Spatial Modeling.** A key design of CFG lies in the disentanglement of 3D spatial modeling across vertical and horizontal directions. We train LEO-VL on the ScanNet subset under two ablated settings: "**w/o z-pos**", where the z-axis position embedding is removed; and "**w/o xyz-pos**", where both z-axis and xy-plane position embeddings are removed. As shown in Fig. 6, both types of position embeddings play critical roles in 3D-VL tasks. Removing the z-axis position embedding leads to a general performance drop, while removing the xy-plane position embedding especially degrades performance on SQA3D. These results validate the effectiveness of our design in 3D spatial modeling.

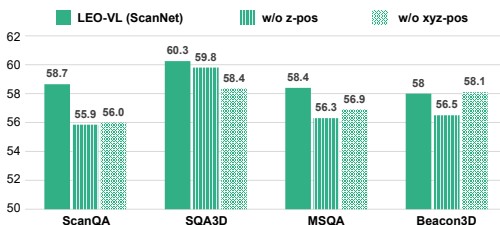 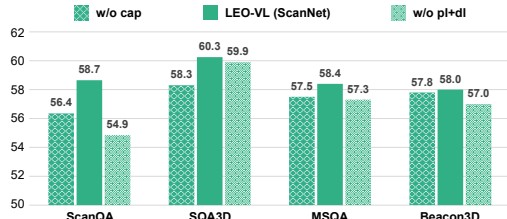

Figure 6: **Ablation on position embeddings.** For more consistent visualization, we use the case-centric metric for Beacon3D, and averaged metrics for others.

Figure 7: **Ablation on task effects.** For more consistent visualization, we use the case-centric metric for Beacon3D, and averaged metrics for others.

Table 4: **Ablation results of scene domain scaling and simplistic QA data.** Benchmarks and metrics follow Table 2, with metrics averaged per benchmark.

| Data | ScanQA | SQA3D | MSQA | Beacon3D | MSQA | MSQA | Avg. |
|------|--------|-------|------|----------|------|------|------|
| ScanNet | **58.7** | 60.3 | 58.4 | 37.9 | 46.1 | 54.7 | 52.7 |
| Scan+3R | 56.5 | 59.8 | 56.0 | 37.5 | 51.0 | 56.1 | 52.8 |
| All | 58.0 | **62.3** | **61.7** | **39.4** | **52.0** | **66.5** | **56.7** |
| Scan+3R | **56.5** | **59.8** | 56.0 | **37.5** | 51.0 | **56.1** | **52.8** |
| +QA | 55.7 | 58.3 | **56.8** | 37.5 | **51.3** | 55.6 | 52.5 |

Table 5: **QA Statistics.** "ΔQA" refers to the extra part of QA data. "Top-15 occupancy" denotes occupancy of top-15 frequent answer templates.

| | Count | Top-15 occupancy |
|--|-------|------------------|
| Scan+3R | 253k | 12.8% |
| ΔQA | 669k | 39.2% |

## 4.3 DATA ANALYSIS

We conduct ablative studies on data to answer the following questions: (1) What is the effect of captioning, planning and dialogue tasks in 3D-VL learning? (2) What is the benefit of scaling across diverse scene domains? (3) What happens when we naively scale up 3D-VL data without considering data quality? (4) How effective is data scaling for LEO-VL with curated data?

**Task Effects.** Based on LEO-VL (ScanNet), we ablate the training data with different task configurations: "**w/o cap**" removes both object captioning and scene captioning tasks, while "**w/o pl+dl**" removes planning and dialogue tasks. As shown in Fig. 7, both task categories contribute meaningfully to the capabilities of LEO-VL. Captioning proves particularly beneficial for SQA3D, whereas planning and dialogue tasks are more influential for ScanQA. We further qualitatively test the "**w/o cap**" model to perform the scene captioning task. The model struggles to generate meaningful scene descriptions, likely due to the lack of training in captioning. These findings underscore the importance of task diversity for both acquiring task-specific skills and enhancing cross-task performance.

**Scene Domains.** To assess the impact of scaling across scene domains, we add a checkpoint trained on ScanNet and 3RScan ("Scan+3R"). As shown in Table 4, "Scan+3R" yields significant improvements on 3RScan and ARKitScenes compared to "ScanNet". Furthermore, "All" shows consistent improvements on all domains compared to "Scan+3R". These results suggest that the performance gains stem from the training in more diverse domains, emphasizing the importance of cross-domain scaling for 3D-VL learning.

**Priority of Quality over Scale.** Based on the results of "Scan+3R", we explore the outcome of naively scaling up 3D-VL data without considering data quality. We expand our default subsets on ScanNet and 3RScan with extra 3D QA data (denoted as "ΔQA") from two sources: (1) LEO (Huang et al., 2024b), which provides 94k QA samples from 3RScan; and (2) MMScan (Lyu et al., 2024), which offers 575k QA samples from ScanNet and 3RScan. The resulting checkpoint is denoted as "+QA". We present QA data statistics, especially the occupancy of top-15 frequent answer templates in Table 5, which indicates a more simplistic answer distribution in "ΔQA". The results in Table 4 reveal that despite substantially more QA data, "+QA" shows degradation in the overall performance compared to "Scan+3R". This shows the harm of scaling with low-quality data, and suggests that prioritizing data quality over scale is critical for effective scaling of 3D-VL learning.

**Consistent Scaling Effects.** We demonstrate that our curated data scheme exhibits consistent scaling effects. Specifically, we take 12.5%, 25%, and 50% of our full data for training, and report the overall performance by averaging metrics across all benchmarks. As shown in Fig. 5, the results exhibit a steady upward trend, demonstrating consistent gains as the data scale increases. This reflects the desired scaling behavior enabled by our curated data scheme, in contrast to the degradation when scaling with low-quality data. Notably, the upward trend shows no sign of saturation, suggesting the potential of further scaling with curated 3D-VL data.

Table 6: **Evaluation results of post-training on SQA3D.** SQA3D represents in-domain evaluation, while Beacon3D and Beacon3D indicate out-of-domain generalization. Metrics are averaged for each benchmark.

| Data | SQA3D | Beacon3D | Beacon3D |
|---|---|---|---|
| Baseline | 61.2 | 26.5 | 21.9 |
| SFT | 62.5 | 27.5 | 22.3 |
| GRPO | 51.6 | 28.1 | 25.0 |
| SceneDPO | **62.7** | **28.4** | **25.2** |

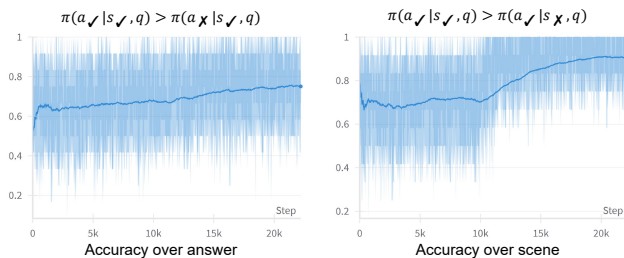

Figure 8: **Accuracy curves during post-training.**

### 4.4 POST-TRAINING

**Settings.** We explore post-training for 3D VLM by continuing to train LEO-VL on SQA3D, comparing SceneDPO with SFT and GRPO. The evaluation includes in-domain test on SQA3D (ScanNet), and out-of-domain test on Beacon3D (3RScan and MultiScan). We report the average of EM and EM-R for SQA3D, and the average of case-centric and object-centric scores for Beacon3D. Training hyperparameters follow the instruction tuning stage. Specific implementations are as follows:

- **SFT.** The implementation is the same as the previous instruction tuning stage.
- **GRPO.** We design an accuracy reward that yields 1 for EM correct, 0.2 for EM-R correct, and 0 otherwise. We have not included format reward due to the lack of cold start Chain-of-Thought (CoT) data. The group size is set to 4, clip range $\epsilon = 0.2$, and KL coefficient $\beta = 0.1$.
- **SceneDPO.** We compile negative answers $a_{\boldsymbol{x}}$ by bootstrapping on model's predictions. We keep the incorrect ones and employ GPT-4o to rephrase the correct ones into negative samples. During post-training, we randomly sample a different scene as the negative scene $s_{\boldsymbol{x}}$. We set $w_a = 0.5$, $w_s = 0.5$, $\beta_a = 0.2$, and $\beta_s = 0.03$.

**Quantitative Results.** As shown in Table 6, all strategies outperform the baseline except GRPO's performance on SQA3D. This implies that rewards may be too sparse to provide effective learning signals for 3D VLM on complex spatial reasoning tasks (*e.g.*, SQA3D). However, GRPO exhibits better out-of-domain performance than SFT, indicating its comparative advantage in enhancing generalizability. In contrast, SceneDPO achieves the best for both in-domain and out-of-domain performances, highlighting its strength as a comprehensive objective. These findings suggest that the potential of GRPO-like post-training may be bottlenecked by the lack of 3D-VL CoT reasoning data, and SceneDPO can be a more practical objective for the post-training of current 3D VLMs.

**Qualitative Results.** Fig. 8 visualizes the dynamic curves of contrast accuracy across answers and scenes during post-training. We observe consistent increasing trends for both accuracies, reflecting an enhancement in the model's ability to discriminate between positive and negative pairs. Notably, the initially low accuracy over scenes reveals the model's limited ability in exploiting scene context for QA, *i.e.*, the issue of visual ignorance, which is a prevalent phenomenon of hallucination in VLMs. These observations resonate with our design motivation and demonstrate the efficacy of SceneDPO for enhancing the robustness of 3D VLMs.

## 5 CONCLUSION

We propose LEO-VL, an advanced 3D VLM that excels at general 3D-VL tasks with natural language outputs. LEO-VL is equipped with an efficient scene representation that features high efficiency, strong perception capability, and simplified 3D spatial modeling. As the representation efficiency unlocks scalability of 3D-VL learning, we curate a comprehensive 3D-VL dataset that spans four real-world indoor domains (*i.e.*, ScanNet, 3RScan, MultiScan, and ARKitScenes) and five tasks, covering captioning, QA, planning, and dialogue. LEO-VL achieves state-of-the-art performance on various 3D QA benchmarks such as SQA3D and Beacon3D. Our extensive ablation studies reveal the importance of diversity in tasks and scene domains, as well as our data curation principle for effective scaling of 3D-VL learning. Furthermore, we demonstrate the advantages of our SceneDPO for the post-training of 3D VLM compared to SFT and GRPO. We hope our experience and findings can benefit future research in 3D VLMs.

ETHICS STATEMENT

This work adheres to the ICLR Code of Ethics. All datasets used in our experiments will be publicly available, and no personally identifiable or sensitive data is involved. Synthetic data generated by LLM is used in a limited and controlled manner to augment training, without introducing privacy, fairness, or safety risks. We believe our work does not pose foreseeable ethical concerns or harmful societal impacts beyond those common to general research in 3D VLMs.

REPRODUCIBILITY STATEMENT

We have provided concrete information to ensure the reproducibility of our work. The model architecture, data scheme, and post-training objective are elaborated in Section 3. Implementation configurations are presented in Sections B and 4. We will release the code and data upon acceptance to facilitate full reproducibility of our results.

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

## A  DISCUSSION

**Limitations.**  Despite strong performance and efficiency, LEO-VL exhibits several limitations. First, the use of the CFG effectively reduces token overhead but may lead to the loss of fine-grained details due to feature pooling, potentially limiting performance on tasks requiring precise understanding. Second, while the CFG's effectiveness is demonstrated on indoor scenes, its generalization to outdoor or dynamic environments remains underexplored. Third, although our data scaling spans diverse real-world scene domains, the potential of leveraging abundant synthetic scenes is not explored yet.

**Broader Impact.**  This work introduces a 3D VLM for scene understanding tasks, offering a step forward in enabling intelligent systems to comprehend and interact with the physical world. The proposed approach holds promise for applications such as assistive robotics, AR/VR interfaces, and digital twin systems for smart environments. Meanwhile, we recognize potential concerns related to data privacy, computational resource consumption, and misuse in surveillance. We encourage the community to pair technical innovation with ethical foresight. And we will release our code and data to support transparent, reproducible, and responsible research in 3D VLMs.

**LLM Usage Statement.**  We use LLM in two ways: (1) to aid with polishing the writing, including improving grammar and coherence of the manuscript; and (2) to generate a portion of training data for our model. The LLM does not contribute to any significant part of this work.

## B  IMPLEMENTATION DETAILS

**Input Sequence.**  Each input to the LLM begins with a system prompt, which is followed by the `Scene Tokens`, derived from the CFG, and then the `Instruction`, *i.e.*, the textual input. For situated tasks, the situation description is included within the `Instruction`, followed by task-specific text such as a question. The model then generates the `Response`, triggered by the generation prompt "assistant".

```
<|im_start|>system
You are a helpful assistant.<|im_end|>
<|im_start|>user
<|vision_start|>{Scene Tokens}<|vision_end|>{Instruction}<|im_end|>
<|im_start|>assistant
{Response}
```

**LLM Hyperparameters.**  We finetune the LLM using LoRA (Hu et al., 2022) for parameter-efficient learning, with a rank of 16, scaling factor $\alpha = 16$, and a dropout rate of 0, following LEO (Huang et al., 2024b). LoRA adapters are applied to all projection matrices: $(W_q, W_k, W_v, W_o)$ in the attention layers and $(W_{gate}, W_{up}, W_{down})$ in the MLP blocks. For inference, we employ beam search with repetition and length penalties. Full hyperparameter settings are provided in Table 7.

**Training Hyperparameters.**  We present the detailed settings in Table 8. The instruction tuning and post-training of LEO-VL are both conducted for 5 epochs.

**GPT-Score.**  We calculate GPT-Score using the Azure OpenAI API with GPT-4o-2024-08-06 as the evaluator. Each score is evaluated based on a triplet consisting of a question, a ground-truth answer, and a predicted answer. We adopt the scoring prompt from MSR3D (Linghu et al., 2024), which includes a system instruction and example contexts to guide the evaluation criteria.

## C  ADDITIONAL RESULTS

**Inference efficiency.**  We evaluate the inference efficiency of LEO-VL (w/ CFG) with (1) LLaVA-3D (Zhu et al., 2024b), (2) Video-3D LLM (Zheng et al., 2025b), and (3) the vanilla VLM backbone of LEO-VL (w/o CFG) (Bai et al., 2025). Metrics are reported on ScanNet scene0050_00 using the same prompt with an NVIDIA 4090 GPU. As shown in Table 9, LEO-VL (w/ CFG) exhibits the highest efficiency, especially in FLOPs and computation time.

Table 7: **Hyperparameters of the LLM.**

| Hyperparameter | Value |
|---|---|
| Training strategy | LoRA |
| Rank | 16 |
| Alpha | 16 |
| Dropout | 0.0 |
| Inference strategy | Beam search |
| Number of beams | 5 |
| Max new tokens | 256 |
| Repetition penalty | 3.0 |
| Length penalty | 1.0 |

Table 8: **Training hyperparameters of LEO-VL.**

| Hyperparameter | Value |
|---|---|
| Optimizer | AdamW |
| Weight decay | 0.05 |
| Betas | [0.9, 0.999] |
| Base learning rate | $3 \times 10^{-5}$ |
| Warmup steps | 400 |
| Type of GPUs | NVIDIA A100/A800 80G |
| Number of GPUs | 8 |
| Batch size per GPU (total) | 2 (16) |
| Gradient accumulation steps | 5 |
| Training precision | bfloat16 |
| Gradient norm | 5.0 |

Table 9: **Inference efficiency.** The metrics are reported on ScanNet scene0050_00 with an NVIDIA 4090 GPU.

| Inference Efficiency | FLOPs (T) | Time (s) | Memory (GB) |
|---|---|---|---|
| LLaVA-3D (Zhu et al., 2024b) | 23.3 | 0.240 | 14.961 |
| Video-3D LLM (Zheng et al., 2025b) | 116.7 | 1.375 | 21.689 |
| Qwen2.5-VL (w/o CFG) (Bai et al., 2025) | 114.1 | 1.122 | 17.947 |
| LEO-VL (w/ CFG) | 13.8 | 0.170 | 14.959 |

Table 10: **Performance on 3D captioning task.** We compare metrics under IoU@0.5 on Scan2Cap val set.

| Model | CIDEr | BLEU-4 | METEOR | ROUGE |
|---|---|---|---|---|
| Scan2Cap (Chen et al., 2021) | 35.2 | 22.4 | 21.4 | 43.5 |
| 3DJCG (Cai et al., 2022) | 49.5 | 31.0 | 24.2 | 50.8 |
| 3D-VisTA (Zhu et al., 2023) | 66.9 | 34.0 | 27.1 | 54.3 |
| Vote2Cap-DETR++ (Chen et al., 2024b) | 74.4 | 37.2 | 26.2 | 53.3 |
| 3DVLP (Zhang et al., 2024a) | 54.4 | 34.1 | **34.3** | 54.3 |
| LEO (Huang et al., 2024b) | 72.4 | 38.2 | 27.9 | 58.1 |
| LL3DA (Chen et al., 2024a) | 65.2 | 36.8 | 26.0 | 55.1 |
| PQ3D (Zhu et al., 2024c) | 80.3 | 36.0 | 29.1 | 57.9 |
| Chat-Scene (Huang et al., 2024a) | 77.2 | 36.3 | 28.0 | 58.1 |
| LLaVA-3D (Zhu et al., 2024b) | 79.2 | 41.1 | 30.2 | 63.4 |
| 3D-LLaVA (Deng et al., 2025) | 78.8 | 36.9 | 27.1 | 57.7 |
| Video-3D LLM (Zheng et al., 2025b) | 80.0 | 40.2 | 28.5 | 61.7 |
| GPT4Scene (Qi et al., 2025) | **86.3** | 40.6 | 28.2 | 59.3 |
| LEO-VL | 77.3 | **43.8** | 30.3 | **64.9** |

**Scan2Cap.** Scan2Cap (Chen et al., 2021) is a benchmark for 3D object captioning, which utilizes the referential text from ScanRefer (Chen et al., 2020) as captions. We have not included Scan2Cap results in the main paper as the captions are limited in linguistic diversity and offer limited insight into a model's descriptive capability. Nonetheless, for completeness, we report the performance of LEO-VL finetuned on Scan2Cap in Table 10. The results show that LEO-VL achieves state-of-the-art performance on Scan2Cap, attaining the highest scores on two out of four evaluation metrics, highlighting its strong capability in object captioning.

**ScanRefer and Nr3D.** ScanRefer (Chen et al., 2020) and Nr3D (Achlioptas et al., 2020) are two benchmarks for 3D object grounding. We finetune LEO-VL on ScanRefer and Nr3D datasets, requiring the model to directly output 3D object bboxes in the format $[x_{center}, y_{center}, z_{center}, l_x, l_y, l_z]$. Similar to prior works (Chen et al., 2024a; Zhu et al., 2024b), we prompt with an anchor pillar in the CFG. The results are shown in Table 11, confirming LEO-VL's compatibility with the grounding task. We present key takeaways below.

- Competitive grounding performance. Despite the inherent difficulty of directly predicting 3D bboxes (Chen et al., 2024a; Zhu et al., 2024a;b), LEO-VL achieves strong performance under Acc@IoU25, confirming the effectiveness of our 3D spatial encoding, despite the vertical conden-

Table 11: **Performance on 3D grounding task.** [*] and [†] indicate results without and with object proposals.

| Model | ScanRefer | | Nr3D | |
|---|---|---|---|---|
| | Acc@IoU25 | Acc@IoU50 | Acc@IoU25 | Acc@IoU50 |
| ReGround3D[*] (Zhu et al., 2024a) | 53.1 | 41.1 | - | - |
| LLaVA-3D[*] (Zhu et al., 2024b) | 50.1 | 42.7 | - | - |
| Chat-Scene[†] (Huang et al., 2024a) | 55.5 | 50.2 | - | - |
| Video-3D LLM[†] (Zheng et al., 2025b) | 57.9 | 51.2 | - | - |
| LEO-VL[*] | 60.0 | 20.6 | 48.3 | 15.1 |
| LEO-VL[†] | 83.7 | 75.2 | 73.0 | 63.3 |

Table 12: **Detailed results on ScanQA val set.**

| Model | CIDEr | BLEU-4 | METEOR | ROUGE | EM | EM-R |
|---|---|---|---|---|---|---|
| *Domain scaling* | | | | | | |
| LEO-VL (ScanNet) | **101.1** | **16.2** | **19.6** | **47.4** | **23.0** | **45.9** |
| LEO-VL (Scan+3R) | 97.2 | 15.7 | 19.0 | 45.7 | 21.9 | 44.3 |
| LEO-VL | 100.4 | 15.5 | 19.5 | 46.9 | 22.6 | 45.3 |
| *Voxel vs. CFG* | | | | | | |
| voxel-full | **101.5** | 15.7 | **20.0** | **48.0** | **23.3** | **46.4** |
| voxel-sample | 96.5 | 15.2 | 19.2 | 45.9 | 21.6 | 43.5 |
| LEO-VL (ScanNet) | 101.1 | **16.2** | 19.6 | 47.4 | 23.0 | 45.9 |
| *Position embedding* | | | | | | |
| w/o xyz-pos | 97.7 | 14.3 | 19.2 | 46.1 | 21.9 | 44.2 |
| w/o z-pos | 97.1 | 14.6 | 19.1 | 46.1 | 21.8 | 44.0 |
| LEO-VL (ScanNet) | **101.1** | **16.2** | **19.6** | **47.4** | **23.0** | **45.9** |
| *Task effects* | | | | | | |
| w/o cap | 97.9 | 14.8 | 19.2 | 46.2 | 22.3 | 44.3 |
| w/o pl+dl | 95.8 | 13.9 | 18.7 | 45.8 | 21.9 | 44.7 |
| LEO-VL (ScanNet) | **101.1** | **16.2** | **19.6** | **47.4** | **23.0** | **45.9** |
| *Quality vs. scale* | | | | | | |
| LEO-VL (Scan+3R) | **97.2** | **15.7** | **19.0** | **45.7** | 21.9 | **44.3** |
| +QA | 96.4 | 14.9 | 18.9 | 45.4 | **22.0** | 43.1 |
| *Data scaling* | | | | | | |
| 12.5% | 79.2 | 10.6 | 15.9 | 39.6 | 19.4 | 38.3 |
| 25% | 86.4 | 11.4 | 17.3 | 42.6 | 20.8 | 41.1 |
| 50% | 94.8 | 13.6 | 18.7 | 44.7 | 21.2 | 42.0 |
| LEO-VL | **100.4** | **15.5** | **19.5** | **46.9** | **22.6** | **45.3** |

sation. However, a relatively weaker Acc@IoU50 suggests a limitation in finer-grained localization, likely due to the compression in representation.

• Improvement with object proposals. Some 3D VLMs (Huang et al., 2024a; Zheng et al., 2025b) leverage Mask3D object proposals (Schult et al., 2023) to simplify the object grounding task as a multi-choice task. We also report results where we retrieve the Mask3D object proposal with the maximum IoU with the model's raw prediction as the final prediction. This approach shows significantly higher accuracies, demonstrating great improvements when aided by object proposals.

**ScanQA and SQA3D.** We present extended results with more metrics for the experiments reported in the main paper. We report the full results for ScanQA in Table 12 and SQA3D in Table 13.

**Beacon3D.** We present detailed results with more metrics in Table 14.

**Analysis on vertical spatial relations.** To probe the model's ability to handle vertical spatial relations, we curate a specialized "vertical-spatial-relation" subset of 34 questions from the Beacon3D ScanNet data, which show patterns like "What is (directly) on/above/under/below ..." We evaluate LEO-VL's performance on this subset, which yields 58.1% accuracy, notably higher than the 41.2%

Table 13: **Detailed results on SQA3D test set.**

| Model | What | Is | How | Can | Which | Others | EM | EM-R |
|---|---|---|---|---|---|---|---|---|
| ***Domain scaling*** | | | | | | | | |
| LEO-VL (ScanNet) | **52.0** | 71.9 | 55.1 | 67.8 | 54.7 | 57.1 | 58.7 | 61.8 |
| LEO-VL (Scan+3R) | 50.2 | 71.8 | 54.0 | **69.2** | 57.8 | 56.5 | 58.3 | 61.3 |
| LEO-VL | 51.5 | **72.9** | **58.9** | 68.9 | **61.8** | **61.5** | **60.8** | **63.7** |
| ***Voxel vs. CFG*** | | | | | | | | |
| voxel-full | 51.8 | 71.0 | **55.5** | 66.9 | 49.6 | **58.1** | 58.1 | 61.1 |
| voxel-sample | 49.0 | 69.8 | 51.4 | **68.0** | **55.8** | 54.6 | 56.6 | 59.7 |
| LEO-VL (ScanNet) | **52.0** | **71.9** | 55.1 | 67.8 | 54.7 | 57.1 | **58.7** | **61.8** |
| ***Position embedding*** | | | | | | | | |
| w/o xyz-pos | 50.5 | 71.6 | 54.2 | 67.2 | 47.3 | 55.5 | 57.0 | 59.7 |
| w/o z-pos | 50.9 | 71.6 | **56.6** | 66.3 | 53.8 | **58.0** | 58.4 | 61.2 |
| LEO-VL (ScanNet) | **52.0** | 71.9 | 55.1 | **67.8** | **54.7** | 57.1 | **58.7** | **61.8** |
| ***Task effects*** | | | | | | | | |
| w/o cap | 50.1 | 70.9 | 54.4 | **68.9** | 46.4 | 55.3 | 56.8 | 59.8 |
| w/o pl+dl | **52.7** | **71.9** | 53.5 | **68.9** | 53.0 | 55.3 | 58.4 | 61.4 |
| LEO-VL (ScanNet) | 52.0 | **71.9** | **55.1** | 67.8 | **54.7** | **57.1** | **58.7** | **61.8** |
| ***Quality vs. scale*** | | | | | | | | |
| LEO-VL (Scan+3R) | **50.2** | 71.8 | **54.0** | **69.2** | **57.8** | **56.5** | **58.3** | **61.3** |
| +QA | 49.4 | **73.2** | 52.5 | 66.3 | 49.6 | 54.8 | 56.7 | 59.8 |
| ***Data scaling*** | | | | | | | | |
| 12.5% | 44.8 | 67.3 | 46.5 | 66.9 | 59.5 | 53.7 | 54.2 | 56.9 |
| 25% | 46.4 | 67.3 | 48.6 | 64.8 | 53.3 | 55.1 | 54.4 | 57.1 |
| 50% | 50.2 | 70.1 | 52.7 | 66.9 | 50.7 | 54.6 | 56.6 | 59.3 |
| LEO-VL | **51.5** | **72.9** | **58.9** | **68.9** | **61.8** | **61.5** | **60.8** | **63.7** |
| ***Post-training on SQA3D*** | | | | | | | | |
| SFT | 53.0 | 73.3 | **55.5** | **68.3** | 65.2 | 60.8 | 61.0 | 64.0 |
| GRPO | 35.5 | 52.0 | 0.0 | 63.0 | 50.1 | 47.7 | 39.9 | 63.3 |
| SceneDPO | **53.1** | **73.5** | 54.4 | 68.0 | **66.4** | **61.3** | **61.1** | **64.2** |

Table 14: **Detailed results on Beacon3D (ScanNet).**

| Model | Class | App. | Geo. | Spa. | Exi. | Overall (case) | Overall (obj.) |
|---|---|---|---|---|---|---|---|
| 3D-VisTA (Zhu et al., 2023) | 28.4 | 35.7 | 41.6 | 48.0 | 55.0 | 43.2 | 7.3 |
| PQ3D (Zhu et al., 2024c) | 37.8 | 45.8 | 32.1 | 19.2 | 44.5 | 35.9 | 4.2 |
| SceneVerse (Jia et al., 2024) | 26.4 | 40.4 | 40.0 | 35.0 | 54.1 | 40.5 | 4.7 |
| LEO (Huang et al., 2024b) | 16.4 | 39.8 | 47.6 | 52.8 | 54.3 | 45.2 | 7.5 |
| Chat-Scene (Huang et al., 2024a) | 30.0 | 42.7 | 50.0 | 53.9 | 62.9 | 49.8 | 10.9 |
| LLaVA-3D (Zhu et al., 2024b) | 35.1 | 66.7 | **62.5** | 54.2 | 62.9 | 59.1 | 19.0 |
| Video-3D LLM (Zheng et al., 2025b) | 40.1 | 64.1 | 60.6 | 55.3 | 64.1 | 59.0 | 17.9 |
| GPT4Scene (Qi et al., 2025) | 38.1 | 59.7 | 59.3 | 52.6 | **66.1** | 57.2 | 17.9 |
| LEO-VL (ScanNet) | **43.0** | 65.8 | 56.7 | 56.6 | 56.3 | 58.0 | 17.7 |
| LEO-VL | 41.2 | **67.4** | 57.0 | **61.0** | 56.7 | **59.5** | **19.2** |

accuracy on the parent question category. This indicates that vertical condensation does not make this question type a particular challenge for LEO-VL. We identify 15 failure cases in this subset and attribute them to three categories, as illustrated in Table 15. We observe that the primary failure mode remains general hallucination/overfitting, rather than confusion on overlapped objects. This demonstrates that the vertical pooling would not be a bottleneck for 3D spatial reasoning.

**SceneDPO ablation.** We present the ablation results of SceneDPO loss terms in Table 16, which confirm that both NLL loss ($\mathcal{L}_{NLL}$) and scene contrast loss ($\mathcal{L}_s$) positively contribute to the final performance. More specifically, we observe that adding the scene contrast loss ($\mathcal{L}_s$) leads to notable improvements in out-of-domain (OOD) generalization (50.4 → 52.8 on 3RScan; and 37.7 → 39.4

Table 15: **The attribution of failure case on vertical-spatial-relation subset.**

| Failure Category | Count | Definition | Example |
|---|---|---|---|
| Hallucination/Overfitting | 7/15 | The model answers with a likely object based on prior knowledge | What is above the couch? → window (actually there is no window) |
| Confusion on Overlapped Objects | 3/15 | The model mistakes an object very close in height within the same pillar for the target | Misidentifying a pillow on a bed as "above the bed" when a picture is the correct answer |
| Grounding/Classification Failure | 5/15 | The model fails to correctly ground or classify the related object instance | Misidentifying an oven as a dishwasher |

Table 16: **Ablation results of SceneDPO loss.** Beacon3D 3RScan and MultiScan indicate OOD evaluation.

| SQA3D Post-training | SQA3D EM | SQA3D EM-R | Beacon3D 3RScan EM-R | Beacon3D MultiScan EM-R |
|---|---|---|---|---|
| SceneDPO | 61.1 | 64.2 | 50.0 | 40.3 |
| - NLL ($\mathcal{L}_{NLL}$) | 59.6 | 62.6 | 52.8 | 39.4 |
| - scene ($\mathcal{L}_s$) | 60.3 | 63.2 | 50.4 | 37.7 |

on MultiScan). The full SceneDPO effectively remedies the weak performance of vanilla DPO, providing notable gains in both in-domain and OOD evaluation results.

## D POSITION EMBEDDING

### D.1 ROTARY POSITION EMBEDDING

**Introduction.** Position embeddings are essential in Transformer-based (Vaswani et al., 2017) architectures, which are inherently permutation-invariant. Traditional absolute position embeddings are simple but exhibit poor generalizability to longer sequences and limited capacity for capturing relative positions. RoPE (Su et al., 2024) addresses these limitations by encoding positions as rotations in a complex plane. This formulation inherently supports relative position encoding and can generalize to sequences of unseen lengths. Due to these advantages, RoPE has been widely adopted in recent models such as LLaMA (Touvron et al., 2023).

**Formulation.** Let $x_p \in \mathbb{R}^d$ denote a feature vector at position $p$. RoPE decomposes $x_p$ into a sequence of 2D subspaces and applies rotation matrices parameterized by channel-specific frequencies:

$$\text{RoPE}(x_p) = R_p x_p,$$

where the rotation matrix $R_p \in \mathbb{R}^{d \times d}$ is block-diagonal, composed of $\frac{d}{2}$ 2D rotation submatrices:

$$
R_p = \begin{pmatrix}
\cos p\theta_1 & -\sin p\theta_1 & 0 & 0 & \cdots & 0 & 0 \\
\sin p\theta_1 & \cos p\theta_1 & 0 & 0 & \cdots & 0 & 0 \\
0 & 0 & \cos p\theta_2 & -\sin p\theta_2 & \cdots & 0 & 0 \\
0 & 0 & \sin p\theta_2 & \cos p\theta_2 & \cdots & 0 & 0 \\
\vdots & \vdots & \vdots & \vdots & \ddots & \vdots & \vdots \\
0 & 0 & 0 & 0 & \cdots & \cos p\theta_{\frac{d}{2}} & -\sin p\theta_{\frac{d}{2}} \\
0 & 0 & 0 & 0 & \cdots & \sin p\theta_{\frac{d}{2}} & \cos p\theta_{\frac{d}{2}}
\end{pmatrix},
$$

with channel-specific frequencies $\theta_i = 10000^{-\frac{2i}{d}}$.

**Height Encoding in 3D Scenes.** We adopt RoPE to encode the voxel heights in 3D scenes considering its advantage in capturing height information during voxel feature pooling. As illustrated in Fig. 9, when two objects swap their heights, RoPE-encoded features can distinguish the configurations, whereas additive position embeddings fail. This property motivates our adoption of RoPE for the encoding of height information.

### D.2 FOURIER FEATURES

**Introduction.** Fourier features (Tancik et al., 2020; Li et al., 2021) focus on encoding multi-dimensional spatial positions, such as pixel positions in an image. Classical methods typically process

Figure 9: **Height encoding with RoPE.** When two objects swap their vertical positions, RoPE preserves the distinction in feature, while additive embeddings cannot.

each dimension independently and then combine via concatenation. However, this falls short in capturing holistic spatial relations such as Euclidean distances, and struggles with generalizing to unbounded ranges of multi-dimensional coordinates. Fourier features overcome these issues by mapping the low-dimensional coordinates into a high-dimensional periodic function space, enabling better modeling of multi-dimensional positions.

**Formulation.** Given an $n$-dim spatial coordinate $p \in \mathbb{R}^n$ and a projection matrix $W \in \mathbb{R}^{\frac{d}{2} \times n}$ (where $d$ is the feature dimension), the Fourier feature $F \in \mathbb{R}^d$ is computed as:

$$F = \left[\cos\left(2\pi W p\right)^{\mathrm{T}} \| \sin\left(2\pi W p\right)^{\mathrm{T}}\right]^{\mathrm{T}},$$

where $W \sim \mathcal{N}(0, 1)$ is initialized from a Gaussian distribution. The resulting feature vector is then scaled by $\frac{1}{\sqrt{d}}$ for normalization and passed through an MLP to generate the final position embedding, which is subsequently added to the original feature vector $x_p$.

# E  DATA STATISTICS

**Additional Simplistic QA Data.** In the main paper, we explore the outcome of naively scaling with low-quality 3D-VL data. The additional QA data is sourced from LEO (Huang et al., 2024b) (94k from 3RScan) and MMScan (Lyu et al., 2024) (575k from ScanNet and 3RScan). This part of data exhibits a simplistic distribution with limited answer diversity. By categorizing answers into placeholder-based templates, we observe a significantly higher proportion of the top-15 frequent answer templates in the additional QA data (39.2% *vs.* 12.8%). The templates and their distribution are visualized in Figs. 10 and 11.

# F  QUALITATIVE EXAMPLES

We present qualitative results of LEO-VL performing various tasks in Fig. 12.

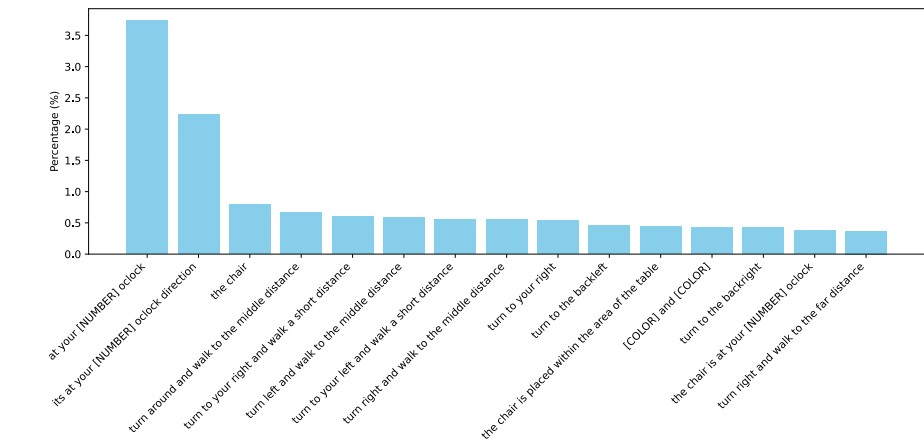

Figure 10: **Top-15 frequent answer templates (12.8%) in our default QA data on ScanNet and 3RScan.**

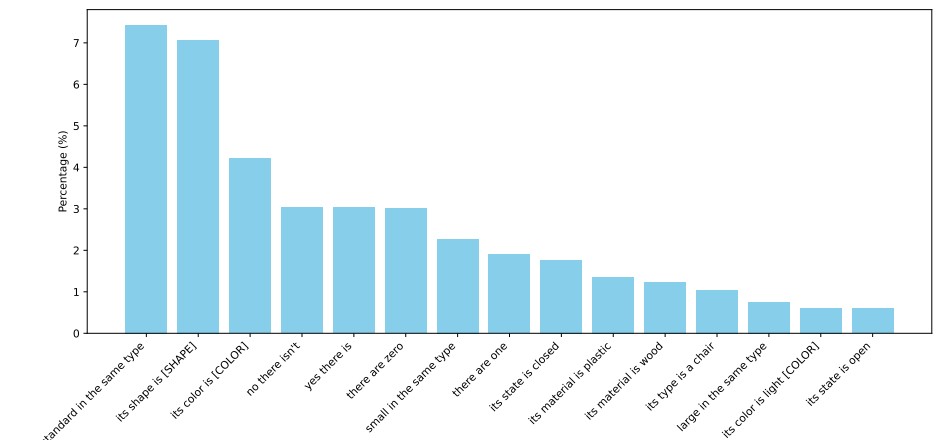

Figure 11: **Top-15 frequent answer templates (39.2%) in the additional QA data on ScanNet and 3RScan.**

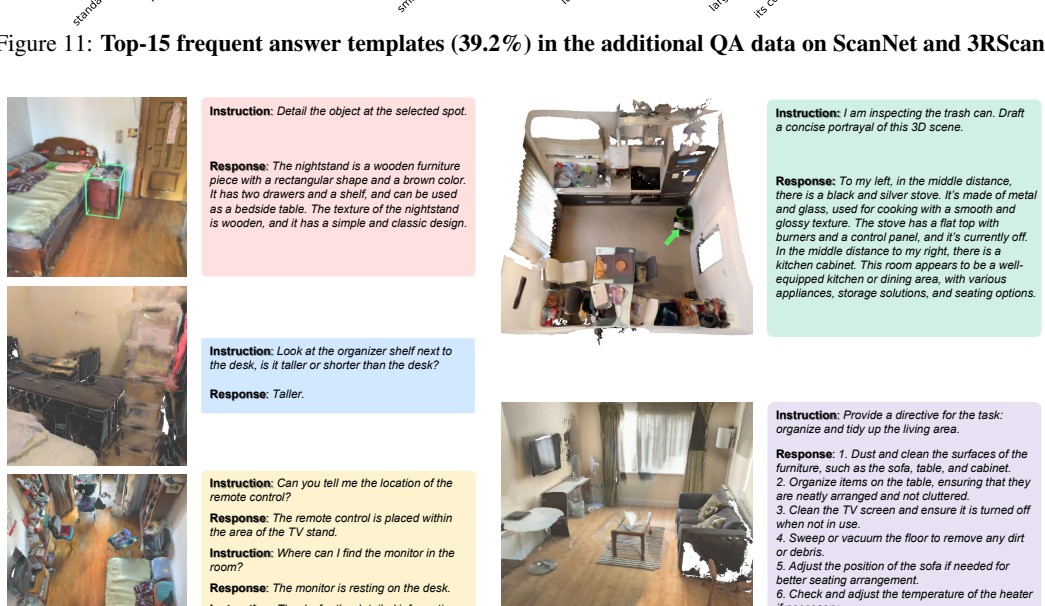

Figure 12: **Qualitative results of LEO-VL on various tasks.** The left column shows object captioning, QA, and dialogue; the right column shows scene captioning and planning.

