# OpenReview forum: "LEO-VL: Efficient Scene Representation for Scalable 3D Vision-Language Learning"
_ICLR.cc/2026/Conference — Submitted to ICLR 2026_

### Official Review · Reviewer_MyZJ · 2025-10-24

**Soundness:** 2
**Presentation:** 3
**Contribution:** 2
**Rating:** 2
**Confidence:** 4

**Summary:**

This paper introduces a 3D Vision-Language Model built upon a condensed feature grid for efficient scene representation. The authors further construct a large-scale 3D Vision-Language instruction-tuning dataset to validate the scalability of their method, along with a reinforcement learning  post-training approach. The proposed method achieves state-of-the-art performance across various 3D understanding benchmarks.

**Strengths:**

1. The paper introduces new datasets for 3D understanding with detailed ablation studies and demonstrates strong scalability.
2. The proposed method achieves leading performance across multiple 3D understanding benchmarks.
3. The idea of post-training a 3D LLM with reinforcement learning is promising.

**Weaknesses:**

1. **Computation overhead:** The idea of projecting image features from 2D foundation models is not novel—3D-LLM has already explored this approach. Compared with image-based 3D LLMs such as LLaVA-3D, the back-projection and voxelization steps introduce additional computation. Although the scene encoding reduces tokens from 3,096 (LLaVA-3D) to 750, stricter comparisons in FLOPs, computation time, and memory usage are needed to validate the claimed efficiency.
2. **Dataset advantage:** Given the significantly larger 3D annotation dataset used, it remains unclear whether the improvement stems mainly from the enhanced 3D encoding or the dataset scale. No controlled ablation with a base model (e.g., LEO) is provided to isolate the dataset’s contribution.
3. **RL method analysis:** The RL approach introduces two additional loss terms with extra hyperparameter tuning, yet no ablation studies are provided to demonstrate the necessity of each term. Is the model sensitive to the hyperparameters mentioned in Lines 456–457?

**Questions:**

1. **DPO pair construction:** How are the DPO pairs constructed—by random sampling from all responses or from a pretrained model?
2. **GRPO results:** Why does GRPO achieve worse in-domain performance compared to the base model while improving generalization?
3. **Memory efficiency:** Does processing and storing the feature grid require additional memory, and if so, how does it affect parallel training performance?

---

> ### Author Response · Authors · 2025-11-25
> **Rebuttal to Reviewer MyZJ (Part 1)**
>
> Thanks for your valuable feedback and constructive comments. We appreciate the positive recognition of our work's strengths, including **data-front contribution**, **leading performance**, and **promising idea in post-training**. We will address the reviewer's concerns as below.
>
> ---
>
> > Weakness 1: Computation overhead: The idea of projecting image features from 2D foundation models is not novel—3D-LLM has already explored this approach. Compared with image-based 3D LLMs such as LLaVA-3D, the back-projection and voxelization steps introduce additional computation. Although the scene encoding reduces tokens from 3,096 (LLaVA-3D) to 750, stricter comparisons in FLOPs, computation time, and memory usage are needed to validate the claimed efficiency.
>
> **`A1:`** Good suggestion! We would like to first clarify that our unique part lies in the **condensed feature grid (CFG) rather than "back-projection and voxelization"**, which pertains to the approach of prior work [1]. To address the reviewer's concerns on computation overhead, we compare the inference efficiency of LEO-VL (w/ CFG) with (1) LLaVA-3D [1], (2) Video-3D LLM [2], and (3) the vanilla VLM backbone Qwen2.5-VL (w/o CFG). Metrics are reported on ScanNet scene0050_00 using the same prompt on an NVIDIA 4090 GPU. The results show that **LEO-VL (w/ CFG) exhibits the highest efficiency, especially in FLOPs and computation time**.
>
> | Inference Efficiency  | FLOPs (T) | Time (s) | Memory (GB) |
> | ------ | ---- | ---- | ---- |
> | LLaVA-3D [1] | 23.3 | 0.240 | 14.961 |
> | Video-3D LLM [2] | 116.7 | 1.375 | 21.689 |
> | Qwen2.5-VL (w/o CFG) | 114.1 | 1.122 | 17.947 |
> | LEO-VL (w/ CFG) | 13.8 | 0.170 | 14.959 |
>
> ---
>
> > Weakness 2: Dataset advantage: Given the significantly larger 3D annotation dataset used, it remains unclear whether the improvement stems mainly from the enhanced 3D encoding or the dataset scale. No controlled ablation with a base model (e.g., LEO) is provided to isolate the dataset’s contribution.
>
> **`A2:`** Good point! We add an experiment of training LEO [3] on the ScanNet subset of LEO-VL data, comparing to the performance of LEO-VL (ScanNet). The results, shown in the table below, clearly demonstrate that **LEO-VL significantly surpasses LEO** across various benchmarks when **trained on the same data**. This isolates the dataset's contribution and showcases the **improvement from the model front**. To further isolate the contribution from **VLM backbone** (*i.e.*, Qwen2.5-VL), we train Qwen2.5-VL on the same data and show that **even with much higher computation overhead, it still underperforms LEO-VL**. This highlights the **improvement from our efficient scene representation**.
>
> | ScanNet | ScanQA (CIDEr) | ScanQA (BLEU-4) | SQA3D (EM) | SQA3D (EM-R) | MSQA (GPT-Score) | Beacon3D (Case) | Beacon3D (Obj.) |
> | ------ | ---- | ---- | ---- | ---- | ---- | ---- | ---- |
> | LEO [3] | 80.3 | 12.2 | 49.4 | 51.8 | 53.4 | 48.4 | 9.4 |
> | Qwen2.5-VL (backbone) | 96.6 | 16.0 | 54.1 | 57.1 | 56.1 | 55.4 | 17.4 |
> | LEO-VL (ScanNet) | 101.1 | 16.2 | 58.7 | 61.8 | 58.4 | 58.0 | 17.7 |
>
> ---
>
> > Weakness 3: RL method analysis: The RL approach introduces two additional loss terms with extra hyperparameter tuning, yet no ablation studies are provided to demonstrate the necessity of each term. Is the model sensitive to the hyperparameters mentioned in Lines 456–457?
>
> **`A3:`** Good suggestion! We present the ablation results of SceneDPO loss terms as below, which confirm that both NLL loss ($L_{NLL}$) and scene contrast loss ($L_s$) positively contribute to the final performance. In addition, we note that removing scene contrast loss $L_s$ would also degrade the model's robustness to text overfitting (i.e., the accuracy improvement in Figure 8 would disappear).
> |  SQA3D Post-training | What |  Is  | How  | Can  |Which |Others|  EM  | EM-R |
> | ------ | ---- | ---- | ---- | ---- | ---- | ---- | ---- | ---- |
> |SceneDPO| **53.1** | 73.5 | 54.4 | 68.0 | 66.4 | **61.3** | **61.1** | **64.2** |
> | - nll ($L_{NLL}$)  | 51.3 | **73.6** | 52.5 | 65.1 | 63.0 | 61.0 | 59.6 | 62.6 |
> | - scene ($L_s$) | 51.4 | 73.2 | **55.1** | **68.9** | **66.7** | 58.5 | 60.3 | 63.2 |
>
> **Hyperparameter sensitivity.** We found that most hyperparameters are not sensitive. The exception is the contrast coefficient for the scene contrast loss, $\beta_s$, which we set to 0.03 by default:
> - Setting $\beta_s$ too high makes the model prone to over-optimization of the scene contrast objective, which risks degrading general QA performance.
> - Setting $\beta_s$ too small results in an insufficient signal to effectively mitigate text overfitting.
>
> We consider this sensitivity to be acceptable given the **inherent and well-documented sensitivity of DPO-like algorithms [4,5]**, which often require careful tuning of hyperparameters to achieve optimal balance. We will include this detailed ablation and sensitivity discussion in the revised paper.

---

> ### Author Response · Authors · 2025-11-25
> **Rebuttal to Reviewer MyZJ (Part 2)**
>
> > Question 1: DPO pair construction: How are the DPO pairs constructed—by random sampling from all responses or from a pretrained model?
>
> **`A4:`** Good question! As introduced in Section 4.4, we use the ground-truth answers as positive samples and build negative samples by bootstrapping on model's predictions. Specifically, incorrect predictions are directly used while correct predictions are rephrased into hard negatives by GPT-4o.
>
> ---
>
> > Question 2: GRPO results: Why does GRPO achieve worse in-domain performance compared to the base model while improving generalization?
>
> **`A5:`** This is an interesting and critical question! Our hypothesis is rooted in the following factors:
> - **Limitation in the spatial reasoning capability of base model.** We conjecture that the deficiency in the base model's spatial reasoning capability weakens the effectiveness of in-domain learning for GRPO, which is an inherent nature of GRPO [6,7,8]. This limitation is compounded by the limited information provided by short answer labels and sparse rewards.
> - **Preserved OOD generalization.** We conjecture that GRPO is less prone to overfitting or "memorizing" the specifics of the training domain, and this failure to over-optimize the in-domain data inadvertently helps preserve better out-of-domain (OOD) generalization [9], as the model retains more general and transferable knowledge.
>
> ---
>
> > Question 3: Memory efficiency: Does processing and storing the feature grid require additional memory, and if so, how does it affect parallel training performance?
>
> **`A6:`** No, the memory usage of feature grid is negligible as it is derived from the voxelized features, which already constitute the primary memory usage for scene representation. This is evidenced by the efficiency analysis table, which shows that LEO-VL consumes slightly less memory than LLaVA-3D [1].
>
> ---
>
> **We sincerely hope our response and commitment to revision have clarified all the raised points. If you have any question, please do not hesitate to ask for further discussion.**
>
> [1] Zhu et al. LLaVA-3D: A Simple yet Effective Pathway to Empowering LMMs with 3D Capabilities. ICCV 2025.
>
> [2] Zheng et al. Video-3D LLM: Learning Position-Aware Video Representation for 3D Scene Understanding. CVPR 2025.
>
> [3] Huang et al. An Embodied Generalist Agent in 3D World. ICML 2024.
>
> [4] Wu et al. β-DPO: Direct Preference Optimization with Dynamic β. NeurIPS 2024.
>
> [5] Cho et al. Rethinking DPO: The Role of Rejected Responses in Preference Misalignment. EMNLP 2025 Findings.
>
> [6] Shao et al. DeepSeekMath: Pushing the Limits of Mathematical Reasoning in Open Language Models. arXiv, 2024.
>
> [7] Zhao et al. Echo Chamber: RL Post-training Amplifies Behaviors Learned in Pretraining. COLM 2025.
>
> [8] Yue et al. Does Reinforcement Learning Really Incentivize Reasoning Capacity in LLMs Beyond the Base Model? NeurIPS 2025.
>
> [9] Tong et al. Delving into RL for Image Generation with CoT: A Study on DPO vs. GRPO. NeurIPS 2025.

---

> ### Comment · Reviewer_MyZJ · 2025-11-26
> **Unresolved Questions**
>
> Thank you for the authors’ response. I still have the following questions that remain unresolved.
>
> **Efficiency Questions(`A1`)**
>
> It is appreciated that the authors provide a detailed FLOPs and memory table. Although LEO-VL uses fewer tokens (Fig. 3), its memory efficiency appears similar to LLaVA-3D. Could the authors provide a more detailed breakdown of speed and memory usage for each component of the proposed method (2D perception and back-projection, voxelization, and VLM generation)?
>
> **Backbone Ablation Questions(`A2`)**
>
> Was the experiment using Qwen2.5-VL as the backbone conducted on LEO or LEO-VL? It seems that most of the performance gain comes from using a stronger VLM backbone compared to LEO.
>
> **DPO Questions(`A4`)**
>
> The authors mentioned that the DPO pairs are constructed using GPT-4o to bootstrap the model’s predictions. Please provide more details about this bootstrapping process. Baselines such as GRPO optimize the model using their own outputs and the ground-truth answers. Is it fair to introduce additional GPT-4o annotations in your DPO method?
>
> ---
>
> I will raise my score if the above questions are adequately addressed.

---

> ### Author Response · Authors · 2025-11-27
> **Reply to Reviewer MyZJ (Part 1)**
>
> Thanks for your prompt reply, we will answer your unresolved questions below.
>
> ---
>
> > 1. Efficiency Questions: It is appreciated that the authors provide a detailed FLOPs and memory table. Although LEO-VL uses fewer tokens (Fig. 3), its memory efficiency appears similar to LLaVA-3D. Could the authors provide a more detailed breakdown of speed and memory usage for each component of the proposed method (2D perception and back-projection, voxelization, and VLM generation)?
>
> **`A1:`** We compile the detailed results for the speed and memory usage of each component of LEO-VL in the table below. Key takeaways are as follows:
> - **LEO-VL vs. LLaVA-3D.** The peak memory usage occurs in 2D perception and back-projection. Since this stage is shared between LEO-VL and LLaVA-3D, their overall memory consumption remains similar.
> - **LLM forward.** We observe **significantly less time spent and memory usage in LLM forward**, compared to the initial 2D perception stage. This highlights the **reduced token overhead in LLM forward, which is enabled by our efficient representation**.
>
> | Component | Time (s) | Memory (GB) |
> | ------ | ---- | ---- |
> | 2D perception and back-projection | 0.095 | 1.139 |
> | Voxelization | 0.001 | ~0 |
> | CFG construction | 0.001 | 0.271 |
> | LLM | 0.069 | 0.453 |
>
> ---
>
> > 2. Backbone Ablation Questions: Was the experiment using Qwen2.5-VL as the backbone conducted on LEO or LEO-VL? It seems that most of the performance gain comes from using a stronger VLM backbone compared to LEO.
>
> **`A2:`** LEO-VL uses Qwen2.5-VL as the backbone while LEO does not. When LEO was proposed, the prevalent practice involved combining dedicated 3D visual encoders and text-modal LLMs [1,2,3]. With the rapid progress of 2D VLMs (multi-modal LLMs), the performance improvement achieved by leveraging strong 2D VLMs has become quite pronounced. And the research community has recently embraced this new approach [4,5,6].
>
> **Indeed, a significant portion of the performance improvement stems from VLM backbones**, a trend also evidenced in the abalation study of LLaVA-3D [4]. **Nonetheless, we achieve notable further improvements beyond the VLM backbone, particularly with much higher efficiency**.
>
> For a more detailed introduction to the evolution of 3D representation and VLMs, which contextualizes our design choice, please refer to the *second paragraph in our Introduction Section* and the *first paragraph in our Related Work Section* of the paper.

---

> ### Author Response · Authors · 2025-11-27
> **Reply to Reviewer MyZJ (Part 2)**
>
> > 3. DPO Questions: The authors mentioned that the DPO pairs are constructed using GPT-4o to bootstrap the model’s predictions. Please provide more details about this bootstrapping process. Baselines such as GRPO optimize the model using their own outputs and the ground-truth answers. Is it fair to introduce additional GPT-4o annotations in your DPO method?
>
> **`A3:`** Thanks for your insightful question! We will clarify the
> bootstrapping process and discuss the fairness, respectively.
>
> **Bootstrapping process.** The bootstrapping process is used to collect negative answers for the requisite of DPO learning. We still adopt the vanilla ground-truth answers as the positive answers. The collection of negative answers is as follows:
> - **Initial inference.** We first use LEO-VL model to infer predictions on the SQA3D training set.
> - **Collect incorrect predictions.** For predictions that are Refined Exact-Match (EM-R) incorrect, we keep them as the negative answers.
> - **Rephrase correct predictions.** For predictions that are EM-R correct, we use GPT-4o to rephrase them into incorrect ones with the following prompt:
>
> ```
> The input is a python list of QA data.
> You are supposed to generate incorrect answers for each question for the use of DPO training.
> This means you should generate the most probable **one** answer except the correct one, with subtle difference to make it hard negative.
> For example, for multi-choice questions, choose the other one except the correct one;
> for open questions, hallucinate the most plausible (one) answer except the correct one.
> The output should be in json format like the original input, with an additional key `negative`.
> Avoid redundant words or symbols, directly start with the output list, ensure the output can be directly parsed with python `eval` function.
> Input: {text_data}
> ```
>
> We examined a subset of the GPT-4o returned data and confirmed the high reliability of generated negatives. Finally, we merge the incorrect predictions with the rephrased ones to form the set of negative answers.
>
> **Fairness.** We respectfully assert that the use of GPT-4o for negative sample generation does not introduce an unfair advantage.
> - **Necessity and common practice for DPO.** DPO inherently requires a defined set of negative labels. Our bootstrapping process is simply to provide this requisite. And leveraging LLMs like GPT-4o to automate this process has been a common practice [7,8].
> - **No positive sample bias.** The GPT-4o annotations do not involve positive answers. Therefore, they do not bias the model towards a "GPT-4o style" correct answer.
> - **Comparison to GRPO.** While the annotation provides a negative answer, GRPO would sample from a larger space of candidate answers, which often covers the positive/negative answers in DPO and even exceeds with more candidate answers.
>
> In summary, our bootstrapping process merely provides a negative examples as the requisite for DPO learning, and does not compromise the fairness of the comparison.
>
> ---
>
> **We sincerely hope our response could resolve your questions. If you have any question, please do not hesitate to let us know.**
>
> [1] Qi et al. Pointnet++: Deep Hierarchical Feature Learning on Point Sets in a Metric Space. NeurIPS 2017.
>
> [2] Hong et al. 3D-LLM: Injecting the 3D World into Large Language Models. NeurIPS 2023.
>
> [3] Xu et al. PointLLM: Empowering Large Language Models to Understand Point Clouds. ECCV 2024.
>
> [4] Zhu et al. LLaVA-3D: A Simple yet Effective Pathway to Empowering LMMs with 3D Capabilities. ICCV 2025.
>
> [5] Zheng et al. Video-3D LLM: Learning Position-Aware Video Representation for 3D Scene Understanding. CVPR 2025.
>
> [6] Qi et al. GPT4Scene: Understand 3D Scenes from Videos with Vision-Language Models. arXiv, 2025.
>
> [7] Li et al. Multi-modal Preference Alignment Remedies Degradation of Visual Instruction Tuning on Language Models. ACL 2024.
>
> [8] Pi et al. Strengthening Multimodal Large Language Model with Bootstrapped Preference Optimization. ECCV 2024.

---

> > ### Comment · Reviewer_MyZJ · 2025-11-27
> >
> > My concerns have been resolved, and I appreciate the authors’ efforts in the revisions. I will raise my score to 4. However, the contribution of this paper mainly lies in refining previous methods (such as back-projecting 2D features to 3D structure, adding more fine-grained losses to DPO and utilizing better base VLM). These improvements generally enhance performance as expected but lack foundational differences and insights compared to prior work.

---

> ### Author Response · Authors · 2025-11-27
> **Reply to Reviewer MyZJ**
>
> Thanks so much for the feedback and for raising score. We really appreciate you confirming that all the technical concerns have been fully resolved!
>
> However, we respectfully disagree that "refining previous methods can undermine the contribution of a work". This standard seems inconsistent with how the broader community evaluates similar high-impact papers:
> - Back-projection of 2D features has been a practical technique since 2022 [1,2,3]. Under your definition, impactful recent works [4,5] would also lack technical contribution, as they adopt and tailor the previous methods and simply integrate them with 3D positional embeddings to achieve huge gains. However, the community widely acknowledges their contributions because they significantly advance the state-of-the-art in 3D VLMs.
> - Similarly, DPO follow-ups [6,7] made clear contributions by applying and adapting DPO to VLM learning through new data collection or loss terms to address specific learning issues. They are considered contributions because they solve problems in new contexts.
>
> Our paper achieves a dual benefit of efficiency and performance in 3D VLM representation, and our targeted DPO modifications directly solve pressing issues like hallucination and visual ignorance in 3D scene reasoning. We sincerely hope the reviewer could reconsider our contribution within the context of these similar, high-profile works, given that there are no remaining technical or clarity issues.
>
> Thanks again for your dedication.
>
> [1] Ha et al. Semantic Abstraction: Open-World 3D Scene Understanding from 2D Vision-Language Models. CoRL 2022.
>
> [2] Peng et al. OpenScene: 3D Scene Understanding with Open Vocabularies. CVPR 2023.
>
> [3] Jatavallabhula et al. ConceptFusion: Open-set Multimodal 3D Mapping. RSS 2023.
>
> [4] Zhu et al. LLaVA-3D: A Simple yet Effective Pathway to Empowering LMMs with 3D Capabilities. ICCV 2025.
>
> [5] Zheng et al. Video-3D LLM: Learning Position-Aware Video Representation for 3D Scene Understanding. CVPR 2025.
>
> [6] Pi et al. Strengthening Multimodal Large Language Model with Bootstrapped Preference Optimization. ECCV 2024.
>
> [7] Wang et al. mDPO: Conditional Preference Optimization for Multimodal Large Language Models. EMNLP 2024.

---

> ### Comment · Reviewer_MyZJ · 2025-11-27
>
> 1. **Your interpretation overextends my point**;  As described in LLaVA-3D: *Unlike these approaches, our LLaVA-3D directly builds on the well-trained 2D LMM with multi-view images as input. Utilizing the 3D position embeddings, it brings the 2D patches within a 3D spatial context to construct 3D Patches. This 3D scene representation enables quick adaption of LLaVA for 3D scene understanding while preserving its strong 2D image understanding ability.*  This direction is **fundamentally different and cannot be *tailored from* prior feature-projection approaches (for which 3D-LLM [1] is a more accurate related work, corresponding to the voxel-based method in your definition)**. Thus, their contributions are widely accepted by the community.
>
> 2. You state that *Scene-DPO directly solve hallucination*. However, the current submission still does not clearly explain how Scene-DPO mitigates hallucination, as no direct description is provided in the main paper. I also agree with reviewer XwcE that the performance gain appears marginal. In addition, the reported baseline and SFT results do not fully match those in Table 11 (e.g., the averaged score of LEO-VL seems to be 62.3 rather than 62.5). I encourage the authors to more clearly emphasize the role of Scene-DPO in the main text and articulate how it integrates with the overall framework, as it currently appears only as a post-training component with limited 3D-VQA benchmark coverage.
>
> 3. Regarding the extended ablation of the DPO loss: in `A3` removing any component leads to worse results than both the baseline and the SFT models. If my understanding is correct, “–scene’’ corresponds to vanilla DPO. This is surprising, given that both vanilla DPO and GRPO underperform on 3D tasks. Additional results, for example on Beacon3D, would help further substantiate the effectiveness of the proposed method.
>
> I am willing to further raise my score if these points are clearly addressed in the revision, and if the related-work section provides a more in-depth analysis of existing 3D-LLMs to strengthen your contribution (e.g. LLaVA-3D, Video-3D LLMs).
>
> [1] 3D-LLM: Injecting the 3D World into Large Language Models

---

> ### Author Response · Authors · 2025-11-28
> **Reply to Reviewer MyZJ (Part 1)**
>
> Thanks for your feedback and constructive comments. We appreciate your willingness to discuss on the discrepancies. We address your questions below.
>
> ### Discussion on LLaVA-3D and Our Distinction
> We first apologize for the misinterpretation. The current disagreement mainly lies in the assessment of our work's contribution. We respectfully argue that *the reviewer's "refining previous methods" comments* may overlook the contributions in both the outcome and technical aspects. Let's clarify these two dimensions below.
> - **Outcome aspect.** We do agree that LLaVA-3D [1] is a widely acknowledged work, whose main impact stems from the outcome that it provides a practical solution to advancing state-of-the-art of 3D VLMs (utilizing 2D LMM for 3D scene understanding as mentioned by the reviewer). Similarly, our approach delivers an equally important direction and practical solution: to improve the efficiency of scene representation for 3D VLMs while preserving the perception capability and strong performance. This is a critical and non-trivial challenge in the field, particularly since existing efforts to enhance efficiency often resulted in sacrificed performance [2,3]. Therefore, we respectfully argue that, like LLaVA-3D, our work delivers a key direction and practical solution essential for the advancement of 3D VLMs.
> - **Technical aspect.** Technically speaking, LLaVA-3D's contribution is injecting 3D position embedding into LLaVA [4]. We never under-estimate such a technical proposal as "refining a previous method" despite the highly overlapped architecture with LLaVA. In contrast, our approach proposes a larger architectural shift that transforms 3D layout into a 2D planar grid via **vertical condensation** with **significantly reduced token overhead**. And we disentangle the 3D spatial modeling across vertical and horizontal directions to simplify and optimize the spatial encoding.
>
> Finally, we would like to once again acknowledge the impact of and inspiration from LLaVA-3D, and we are honored to follow prior works like LLaVA-3D to advance the community as they did. We respectfully argue that our work should not be characterized as simply "refining previous methods", as we deliver a critial direction and practical solution (novel technical design) regarding the efficient scene representation for 3D VLMs.

---

> ### Author Response · Authors · 2025-11-28
> **Reply to Reviewer MyZJ (Part 2)**
>
> ### Discussion on SceneDPO
> We will address your question per point as below.
>
> ---
>
> > You state that Scene-DPO directly solve hallucination.
>
> We first apologize for the misstatement of "solve hallucination." Our more precise claim is that we explore a practical method to mitigate hallucination.
>
> ---
>
> > However, the current submission still does not clearly explain how Scene-DPO mitigates hallucination, as no direct description is provided in the main paper.
>
> We directly address the "visual ignorance" issue, a straightforward phenomenon of hallucination that is prevalent in current VLMs [5,6,7,8].
> - **Definition.** Visual ignorance is the model's tendency to answer a question while disregarding the visual context, *e.g.*, when queried by "what color is the chair?", the model overfits to answer with "brown" regardless of the specific scene.
> - **Remedy by scene contrasting objective ($\mathcal{L}_{s}$).** Our SceneDPO introduces a loss term $\mathcal{L}_{s}$ that explicitly enforces **scene-grounding** by contrasting the model's prediction on the **positive scene ($s_✓$)** with an **irrelevant, negative scene ($s_✗$)**. This term "discourages the model from predicting the current answer when conditioned on irrelevant scenes". This avoids generating answers that are not visually grounded and thus mitigates hallucination.
> - **Empirical validation.** The qualitative results (Figure 8) confirm the effectiveness in mitigating visual ignorance issue, which shows that the model initially has a **low "accuracy over scenes,"** which reflects this **visual ignorance**. During post-training, this accuracy **consistently increases**, demonstrating its efficacy in enhancing the model's ability to exploit scene context.
>
> ---
>
> > I also agree with reviewer XwcE that the performance gain appears marginal.
>
> We acknowledge that the performance gain on the in-domain task appears not significant. However, we clarify that SceneDPO has more significant results in mitigating hallucination and improving out-of-domain (OOD) generalization.
>
> We also provide a potential explanation for the relatively small in-domain margin: a bottleneck in the base model's spatial understanding capabilities hinders SceneDPO from achieving a larger, more distinguishable performance gap over SFT.
>
> ---
>
> > In addition, the reported baseline and SFT results do not fully match those in Table 11 (e.g., the averaged score of LEO-VL seems to be 62.3 rather than 62.5).
>
> Good question! The difference is because the starting checkpoint for post-training was an older one, different from the final checkpoint reported in overall results (Table 2). These two checkpoints were trained in the same way except different batch sizes, which accounts for the minor performance variance.
>
> ---
>
> > I encourage the authors to more clearly emphasize the role of Scene-DPO in the main text and articulate how it integrates with the overall framework, as it currently appears only as a post-training component with limited 3D-VQA benchmark coverage.
>
> We agree with the reviewer that the current framing of SceneDPO is somewhat imbalanced. Actually, our initial intention is to introduce it as a minor contribution in aiding the post-training of 3D VLMs. So we did not over-emphasize its role, preventing it from being overweighted in the manuscript. We will properly frame its role and compile more complete results.

---

> ### Author Response · Authors · 2025-11-28
> **Reply to Reviewer MyZJ (Part 3)**
>
> ### Additional SceneDPO Ablation
>
> We appreciate your thorough review of our additional ablation results.
>
> ---
>
> > removing any component leads to worse results than both the baseline and the SFT models... This is surprising, given that both vanilla DPO and GRPO underperform on 3D tasks
>
> We agree that the results (vanilla DPO and GRPO underperform on 3D tasks) ware intially surprising. However, this finding is **strong empirical evidence** validating the necessity and contribution of our SceneDPO design. It suggests that, in the 3D-VL domain, simply enforcing answer preference (vanilla DPO) without simultaneously enforcing **scene grounding** leads to less robust model. This outcome underscores that SceneDPO is a crucial specialization of DPO for the post-training of 3D VLMs.
>
> ---
>
> > Additional results, for example on Beacon3D, would help further substantiate the effectiveness of the proposed method.
>
> We did conduct OOD evaluation on Beacon3D, but had not reported the results since the specific checkpoints were missing. Here we can only provide the EM-R metric (accessed through wandb logs), which empirically aligns most closely with the GPT-Score. We integrate the OOD results with in-domain results of SQA3D below.
>
> | SQA3D Post-training | SQA3D EM | SQA3D EM-R | Beacon3D 3RScan EM-R | Beacon3D MultiScan EM-R |
> | ------ | ---- | ---- | ---- | ---- |
> |SceneDPO| **61.1** | **64.2** | 50.0 | **40.3** |
> | - nll ($L_{NLL}$)  | 59.6 | 62.6 | **52.8** | 39.4 |
> | - scene ($L_s$) | 60.3 | 63.2 | 50.4 | 37.7 |
>
> We report two takeaways:
> - Adding the scene contrast loss ($\mathcal{L}_{s}$) leads to notable improvements in OOD generalization ($50.4 \rightarrow 52.8$ on 3RScan; and $37.7 \rightarrow 39.4$ on MultiScan).
> - The full SceneDPO effectively remedies the weak performance of vanilla DPO, providing notable gains in both in-domain and OOD evaluation results.
>
> ### Final Remarks
>
> Thanks again for your thorough review and your engagement in this discussion. We agree that a more precise articulation of our approach is necessary. We look forward to receiving your feedback regarding the convergence of our discussion on these key discrepancies, and we will proceed to revise the manuscript based on your confirmation.
>
> [1] Zhu et al. LLaVA-3D: A Simple yet Effective Pathway to Empowering LMMs with 3D Capabilities. ICCV 2025.
>
> [2] Huang et al. Zero-shot 3D Question Answering via Voxel-based Dynamic Token Compression. CVPR 2025.
>
> [3] Zhang et al. AdaToken-3D: Dynamic Spatial Gating for Efficient 3D Large Multimodal-Models Reasoning. arXiv, 2025.
>
> [4] Liu et al. Visual Instruction Tuning. NeurIPS 2023.
>
> [5] Wang et al. mDPO: Conditional Preference Optimization for Multimodal Large Language Models. EMNLP 2024.
>
> [6] Xie et al. V-DPO: Mitigating Hallucination in Large Vision Language Models via Vision-Guided Direct Preference Optimization. EMNLP 2024 Findings.
>
> [7] Huang et al. Unveiling the Mist over 3D Vision-Language Understanding: Object-centric Evaluation with Chain-of-Analysis. CVPR 2025.
>
> [8] Brown et al. Benchmark Designers Should "Train on the Test Set" to Expose Exploitable Non-Visual Shortcuts. arXiv, 2025.

---

### Official Review · Reviewer_YEFT · 2025-10-28

**Soundness:** 3
**Presentation:** 3
**Contribution:** 2
**Rating:** 4
**Confidence:** 4

**Summary:**

The paper introduces LEO-VL, a 3D vision-language model that leverages a novel, efficient scene representation called condensed feature grid (CFG) to reduce token overhead without sacrificing perception performance. Trained on 700k 3D-VL samples across diverse domains and tasks, LEO-VL incorporates SceneDPO, a contrastive post-training method to further enhance robustness. The model achieves state-of-the-art results on several 3D question answering benchmarks, demonstrating improved efficiency, scalability, and robustness over prior approaches.

**Strengths:**

1. The overall writing of the paper is clear and easy to follow.

2. The proposed token reduction method achieves a good trade-off between performance and efficiency.

3. The authors conduct comprehensive experiments involving various post-training methods and effectively demonstrate the effectiveness of SceneDPO.

**Weaknesses:**

1. Recent relevant works, such as VG-LLM [1] and 3DRS [2], have not been cited or compared in the paper. Including a discussion or comparison with these approaches would strengthen the related work section and contextualize the contributions of this work.

2. The CFG method compresses height information, which may cause the model to lose fine-grained details of layered objects (e.g., a pillow on a bed or a pot on a table). This limitation could impact the model's ability to accurately represent and distinguish between objects in complex scenes. Furthermore, the proposed token compression method might be incompatible with 3D localization tasks, since it leads to a loss of detailed spatial information.

3. The GRPO algorithm used for comparison is not fully implemented, as it lacks both the code-start stage and the format reward component. Additionally, the paper does not clearly explain how rewards are designed for non-verifiable tasks, such as captioning.

References:

[1] Learning from Videos for 3D World: Enhancing MLLMs with 3D Vision Geometry Priors. NeurIPS 2025.

[2] MLLMs Need 3D-Aware Representation Supervision for Scene Understanding. NeurIPS 2025.

**Questions:**

1. Please properly include recent papers related to this work.

2. Please elaborate how the GRPO algorithm is designed in this paper.

---

> ### Author Response · Authors · 2025-11-25
> **Rebuttal to Reviewer YEFT (Part 1)**
>
> Thanks for your valuable feedback and constructive comments. We appreciate the positive recognition of our work's strengths, including **clear and easy-to-follow writing**, **efficient scene representation**, **comprehensive experiments**, and **effective post-training**. We will address the reviewer's concerns as below.
>
> ---
>
> > Weakness 1: Recent relevant works, such as VG-LLM [1] and 3DRS [2], have not been cited or compared in the paper. Including a discussion or comparison with these approaches would strengthen the related work section and contextualize the contributions of this work.
> >
> > Question 1: Please properly include recent papers related to this work.
>
> **`A1:`** Thanks for pointing out! VG-LLM [1] and 3D-RS [2] are excellent concurrent works that improve the 3D-VL understanding capabilities of video-based VLMs. We will add reference to these works in the updated manuscript and highlight the difference in scene representation.

---

> > ### Comment · Reviewer_YEFT · 2025-11-28
> >
> > Thank you for the detailed rebuttal. Overall, I am satisfied with the authors’ responses, particularly regarding the impressive 3D localization capabilities of LEO-VL.
> >
> > I would also like to clarify that VG-LLM and 3DRS are not concurrent works with LEO-VL; VG-LLM and 3DRS were posted on arXiv on May 30th and June 2nd, respectively—approximately three months before the ICLR submission deadline.
> >
> > Given the current submission and the authors’ clarifications, I am inclined to accept this paper and will raise my score to 8.

---

> > > ### Comment · Reviewer_YEFT · 2025-11-28
> > >
> > > However, the edit button is currently unavailable, so I am unable to update my review at this time.
> > >
> > > I will revise my review once editing becomes available again.

---

> ### Author Response · Authors · 2025-11-25
> **Rebuttal to Reviewer YEFT (Part 2)**
>
> > Weakness 2: The CFG method compresses height information, which may cause the model to lose fine-grained details of layered objects (e.g., a pillow on a bed or a pot on a table). This limitation could impact the model's ability to accurately represent and distinguish between objects in complex scenes. Furthermore, the proposed token compression method might be incompatible with 3D localization tasks, since it leads to a loss of detailed spatial information.
>
> **`A2:`** Thanks for your feedback! We will address the reviewer's concerns on (1) the analysis on vertical spatial relations and (2) 3D localization task, respectively.
>
> ### Section 1: Analysis on Vertical Spatial Relations
>
> To probe the model's ability to handle vertical spatial relations, we curate a specialized "vertical-spatial-relation" subset of 34 questions from the Beacon3D ScanNet data, which show patterns like "*What is (directly) on/above/under/below ...*" We evaluate LEO-VL's performance and analyze the failure cases.
>
> **Vertical spatial relation is not a major challenge.** LEO-VL achieves 58.1% accuracy on this specialized subset, which is notably higher than the 41.2% accuracy on the parent question category. This indicates that vertical condensation does not make this question type a particular challenge for LEO-VL.
>
> **Failure case analysis.** Among the 15 identified failure cases in this subset, the vertical pooling is not the dominant cause of error. Failures can be attributed to three categories:
>
> | Failure category | Count | Definition | Example |
> | ------ | ---- | ---- | ---- |
> | Hallucination/Overfitting | 7/15 | The model answers with a likely object based on prior knowledge | What is above the couch? $\to$ window (actually there is no window) |
> | Confusion on Overlapped Objects | 3/15 | The model mistakes an object very close in height within the same pillar for the target | Misidentifying a pillow on a bed as "above the bed" when a picture is the correct answer |
> | Grounding/Classification Failure | 5/15 | The model fails to correctly ground or classify the related object instance | Misidentifying an oven as a dishwasher |
>
> **Ablation of RoPE embedding.** The baseline model without RoPE embedding scores 53.7% accuracy on the "vertical-spatial-relation" subset, compared to the 58.1% accuracy for the model with RoPE embedding. For more complete ablation results on RoPE embedding, we refer the reviewer to our paper (Figure 6).
>
> We will include this detailed analysis in the revised appendix. In summary, we underscore two takeaways:
> - **Vertical-spatial-relation questions do not pose a unique or major bottleneck** for LEO-VL, and the primary failure mode remains **general hallucination/overfitting**, not the vertical pooling itself.
> - RoPE is indeed **beneficial for improving spatial understanding along the vertical axis and handling the potential ambiguity**.
>
>
> ### Section 2: 3D Localization (Grounding) Task
>
> We finetune LEO-VL on ScanRefer and Nr3D datasets, requiring the model to directly output 3D object bboxes in the format $[x_{center}, y_{center}, z_{center}, l_x, l_y, l_z]$, similar to prior works [3,4]. The results confirm LEO-VL's compatibility and we present key takeaways as below.
> - **Competitive grounding performance.** Despite the inherent difficulty of directly predicting 3D bboxes [3,4,5], LEO-VL achieves 60% Acc@IoU25, confirming the effectiveness of our 3D spatial encoding, despite the vertical condensation. However, a relatively weaker Acc@IoU50 suggests a limitation in finer-grained localization, likely due to the compression in representation.
> - **Improvement with object proposals.** Some 3D VLMs [6,7] leverage object proposals (*e.g.*, Mask3D [8]) to simplify the object grounding task as a multi-choice task. We also report results where we retrieve the Mask3D object proposal with the maximum IoU with the model's raw prediction as the final prediction. This approach shows significantly higher accuracies, demonstrating great improvements when aided by object proposals.
>
> | Grounding | ScanRefer (Acc@IoU25) | ScanRefer (Acc@IoU50) | Nr3D (Acc@IoU25) | Nr3D (Acc@IoU50) |
> | ------ | ---- | ---- | ---- | ---- |
> | ReGround3D (w/o proposal) [5] | 53.1 | 41.1 | - | - |
> | LLaVA-3D (w/o proposal) [3] | 50.1 | 42.7 | - | - |
> | Chat-Scene (w/ proposal) [6] | 55.5 | 50.2 | - | - |
> | Video-3D LLM (w/ proposal) [7] | 57.9 | 51.2 | - | - |
> | LEO-VL (w/o proposal) | 60.0 | 20.6 | 48.3 | 15.1 |
> | LEO-VL (w/ proposal) | 83.7 | 75.2 | 73.0 | 63.3 |

---

> ### Author Response · Authors · 2025-11-25
> **Rebuttal to Reviewer YEFT (Part 3)**
>
> > Weakness 3: The GRPO algorithm used for comparison is not fully implemented, as it lacks both the code-start stage and the format reward component. Additionally, the paper does not clearly explain how rewards are designed for non-verifiable tasks, such as captioning.
> >
> > Question 2: Please elaborate how the GRPO algorithm is designed in this paper.
>
> **`A3:`** Thanks for your feedback! We will clarify on (1) cold-start and format reward, and (2) our implementation of GRPO, respectively.
>
> **Cold-start and format reward.** We did not include the cold-start stage primarily because no suitable Chain-of-Thought (CoT) data is currently available for our tasks. Furthermore, incorporating extra CoT data for GRPO would introduce an unfair comparison against the SFT and SceneDPO baselines, which are trained only on the QA pairs. And the format reward was excluded because we empirically found that simply adding this reward without the cold-start stage (like R1-Zero) yielded no improvement over our current setting.
>
> **Our implementation of GRPO.** We have not trained on non-verifiable tasks like captioning, where the reward design is out of the scope of this paper. We conducted the post-training on the SQA3D dataset. Our implementation of GRPO is elaborated in Section 4.4 of the manuscript:
> - Reward design (accuracy reward): The reward function is based on answer correctness, a verifiable metric for the SQA3D task.
>     - Reward = 1 for Exact-Match (EM) correct answers.
>     - Reward = 0.2 for Refined Exact-Match (EM-R) correct answers.
>     - Reward = 0 otherwise.
> - Hyperparameters:
>     - Group size $G=4$
>     - Clip range $\epsilon=0.2$
>     - KL coefficient $\beta=0.1$
>
> ---
>
> **We sincerely hope our response and commitment to revision have clarified all the raised points. If you have any questions, please do not hesitate to ask for further discussion.**
>
> [1] Zheng et al. Learning from Videos for 3D World: Enhancing MLLMs with 3D Vision Geometry Priors. NeurIPS 2025.
>
> [2] Huang et al. MLLMs Need 3D-Aware Representation Supervision for Scene Understanding. NeurIPS 2025.
>
> [3] Zhu et al. LLaVA-3D: A Simple yet Effective Pathway to Empowering LMMs with 3D Capabilities. ICCV 2025.
>
> [4] Chen et al. LL3DA: Visual Interactive Instruction Tuning for Omni-3D Understanding, Reasoning, and Planning. CVPR 2024.
>
> [5] Zhu et al. ScanReason: Empowering 3D Visual Grounding with Reasoning Capabilities. ECCV 2024.
>
> [6] Huang et al. Chat-Scene: Bridging 3D Scene and Large Language Models with Object Identifiers. NeurIPS 2024.
>
> [7] Zheng et al. Video-3D LLM: Learning Position-Aware Video Representation for 3D Scene Understanding. CVPR 2025.
>
> [8] Schult et al. Mask3D: Mask Transformer for 3D Semantic Instance Segmentation. ICRA 2023.

---

> ### Author Response · Authors · 2025-11-28
> **Reply to Reviewer YEFT**
>
> Thanks for your positive feedback. We are glad that we could address your concerns, particularly for acknowledging the impressive 3D localization capabilities of LEO-VL. We agree that VG-LLM and 3DRS precede our ICLR submission date. We will ensure both are appropriately cited and discussed as related work in the final manuscript.

---

### Official Review · Reviewer_DVCP · 2025-10-30

**Soundness:** 3
**Presentation:** 3
**Contribution:** 3
**Rating:** 6
**Confidence:** 4

**Summary:**

This work addresses the performance vs. efficiency challenge in 3D Vision-Language Models (VLMs). Current methods often have high token costs, which hinders scalable training. This paper proposes LEO-VL, a VLM using a "Condensed Feature Grid" (CFG). The CFG is created by back-projecting 2D features from multi-view RGB-D inputs into 3D voxels, encoding height with RoPE, and then condensing all voxels in a vertical pillar into a single token.

This CFG representation reduces token count (to 750) while retaining spatial information, which allows for larger-scale training. The authors use a 700k sample dataset, noting that data quality is more beneficial than naively scaling with low-quality data. The paper also introduces SceneDPO, a post-training objective to improve model robustness by contrasting both answers and scenes. LEO-VL achieves strong results on 3D QA benchmarks (SQA3D, MSQA, Beacon3D) with high efficiency.

**Strengths:**

1. The Condensed Feature Grid is an elegant and effective contribution. It achieves a ~3x token reduction (33% compression rate) compared to the raw voxel grid, appearing to improve performance by preserving spatial information.
2. The analysis in Section 4.3 demonstrating that scaling with 669k of simplistic, low-quality QA data degrades performance is an important contribution. This focus on "quality over scale" is a useful finding for the field.
3. The model achieves new SOTA performance on multiple challenging 3D QA benchmarks, validating the effectiveness of the overall approach.

**Weaknesses:**

1. While the model is trained on five tasks, the primary evaluation is almost exclusively on QA. The model's proficiency at other tasks (grounding, captioning) is not as rigorously benchmarked.
2. The vertical condensation (pooling) inherently loses information about the vertical distribution of features within a single (x, y) pillar. The paper argues RoPE encoding of height mitigates this, but there is no analysis of failure cases. For example, it's unclear if the model can distinguish a cup on a table from a lamp above the table if they fall in the same pillar.
3. The work is entirely focused on indoor scenes. The scalability and effectiveness of the CFG representation for more complex, large-scale outdoor environments remain unexplored.

**Questions:**

1. Regarding the vertical condensation in CFG: Could you provide a qualitative failure analysis? Are there specific types of spatial questions (e.g., "What is directly on top of object A?" vs. "What is above object A?") where the pooling of an entire vertical pillar into one token causes the model to fail? How well does the RoPE height embedding handle ambiguity for multiple, distinct objects stacked vertically?
2. The "quality over scale" finding is excellent. The paper uses "top-15 answer template occupancy" as a metric for low quality. Could you elaborate on the curation process for the final 700k dataset?

---

> ### Author Response · Authors · 2025-11-25
> **Rebuttal to Reviewer DVCP (Part 1)**
>
> Thanks for your valuable feedback and constructive comments. We appreciate the positive recognition of our work's strengths, including **elegant and effective scene representation**, **useful analysis**, and **state-of-the-art performance**. We will address the reviewer's concerns as below.
>
> ---
>
> > Weakness 1: While the model is trained on five tasks, the primary evaluation is almost exclusively on QA. The model's proficiency at other tasks (grounding, captioning) is not as rigorously benchmarked.
>
> **`A1:`** Good suggestion! We refer the reviewer to appendix (Table 9) for captioning results, where LEO-VL also achieves state-of-the-art performance in object captioning.
>
> For the grounding task, we finetune LEO-VL on ScanRefer and Nr3D datasets, requiring the model to directly output 3D object bboxes in the format $[x_{center}, y_{center}, z_{center}, l_x, l_y, l_z]$, similar to prior works [1,2]. The results confirm LEO-VL's compatibility and we present key takeaways as below.
> - **Competitive grounding performance.** Despite the inherent difficulty of directly predicting 3D bboxes [1,2,3], LEO-VL achieves 60% Acc@IoU25, confirming the effectiveness of our 3D spatial encoding, despite the vertical condensation. However, a relatively weaker Acc@IoU50 suggests a limitation in finer-grained localization, likely due to the compression in representation.
> - **Improvement with object proposals.** Some 3D VLMs [4,5] leverage object proposals (*e.g.*, Mask3D [6]) to simplify the object grounding task as a multi-choice task. We also report results where we retrieve the Mask3D object proposal with the maximum IoU with the model's raw prediction as the final prediction. This approach shows significantly higher accuracies, demonstrating great improvements when aided by object proposals.
>
> | Grounding | ScanRefer (Acc@IoU25) | ScanRefer (Acc@IoU50) | Nr3D (Acc@IoU25) | Nr3D (Acc@IoU50) |
> | ------ | ---- | ---- | ---- | ---- |
> | ReGround3D (w/o proposal) [3] | 53.1 | 41.1 | - | - |
> | LLaVA-3D (w/o proposal) [1] | 50.1 | 42.7 | - | - |
> | Chat-Scene (w/ proposal) [4] | 55.5 | 50.2 | - | - |
> | Video-3D LLM (w/ proposal) [5] | 57.9 | 51.2 | - | - |
> | LEO-VL (w/o proposal) | 60.0 | 20.6 | 48.3 | 15.1 |
> | LEO-VL (w/ proposal) | 83.7 | 75.2 | 73.0 | 63.3 |

---

> ### Author Response · Authors · 2025-11-25
> **Rebuttal to Reviewer DVCP (Part 2)**
>
> > Weakness 2: The vertical condensation (pooling) inherently loses information about the vertical distribution of features within a single (x, y) pillar. The paper argues RoPE encoding of height mitigates this, but there is no analysis of failure cases. For example, it's unclear if the model can distinguish a cup on a table from a lamp above the table if they fall in the same pillar.
> >
> > Question 1: Regarding the vertical condensation in CFG: Could you provide a qualitative failure analysis? Are there specific types of spatial questions (e.g., "What is directly on top of object A?" vs. "What is above object A?") where the pooling of an entire vertical pillar into one token causes the model to fail? How well does the RoPE height embedding handle ambiguity for multiple, distinct objects stacked vertically?
>
> **`A2:`** Thanks for your insightful comments! To probe the model's ability to handle vertical spatial relations, we curate a specialized "vertical-spatial-relation" subset of 34 questions from the Beacon3D ScanNet data, which show patterns like "*What is (directly) on/above/under/below ...*" We evaluate LEO-VL's performance and analyze the failure cases.
>
> **Vertical spatial relation is not a major challenge.** LEO-VL achieves 58.1% accuracy on this specialized subset, which is notably higher than the 41.2% accuracy on the parent question category. This indicates that vertical condensation does not make this question type a particular challenge for LEO-VL.
>
> **Failure case analysis.** Among the 15 identified failure cases in this subset, the vertical pooling is not the dominant cause of error. Failures can be attributed to three categories:
>
> | Failure category | Count | Definition | Example |
> | ------ | ---- | ---- | ---- |
> | Hallucination/Overfitting | 7/15 | The model answers with a likely object based on prior knowledge | What is above the couch? $\to$ window (actually there is no window) |
> | Confusion on Overlapped Objects | 3/15 | The model mistakes an object very close in height within the same pillar for the target | Misidentifying a pillow on a bed as "above the bed" when a picture is the correct answer |
> | Grounding/Classification Failure | 5/15 | The model fails to correctly ground or classify the related object instance | Misidentifying an oven as a dishwasher |
>
> **Ablation of RoPE embedding.** The baseline model without RoPE embedding scores 53.7% accuracy on the "vertical-spatial-relation" subset, compared to the 58.1% accuracy for the model with RoPE embedding. For more complete ablation results on RoPE embedding, we refer the reviewer to our paper (Figure 6).
>
> We will include this detailed analysis in the revised appendix. In summary, we underscore two takeaways:
> - **Vertical-spatial-relation questions do not pose a unique or major bottleneck** for LEO-VL, and the primary failure mode remains **general hallucination/overfitting**, not the vertical pooling itself.
> - RoPE is indeed **beneficial for improving spatial understanding along the vertical axis and handling the potential ambiguity**.
>
> ---
>
> > Weakness 3: The work is entirely focused on indoor scenes. The scalability and effectiveness of the CFG representation for more complex, large-scale outdoor environments remain unexplored.
>
> **`A3:`** Yes, as acknowledged in the Limitations section (Appendix A), adapting our framework to outdoor environments is under-explored and is reserved for future work. However, we believe this does not diminish the novelty and technical contributions demonstrated within the scope of 3D indoor scenes.
>
> ---
>
> > Question 2: The "quality over scale" finding is excellent. The paper uses "top-15 answer template occupancy" as a metric for low quality. Could you elaborate on the curation process for the final 700k dataset?
>
> **`A4:`** Thanks for recognizing our findings in data quality. For the curation process, we mainly discard QA data that exhibits overly simplistic linguistic patterns, including 1) MMScan QA data on ScanNet and 3RScan, 2) LEO-1 QA data on 3RScan, and 3) our newly generated QA data on MultiScan.
>
> ---
>
> **Thanks again for your valuable feedback. If you have any further questions, feel free to let us know.**
>
> [1] Zhu et al. LLaVA-3D: A Simple yet Effective Pathway to Empowering LMMs with 3D Capabilities. ICCV 2025.
>
> [2] Chen et al. LL3DA: Visual Interactive Instruction Tuning for Omni-3D Understanding, Reasoning, and Planning. CVPR 2024.
>
> [3] Zhu et al. ScanReason: Empowering 3D Visual Grounding with Reasoning Capabilities. ECCV 2024.
>
> [4] Huang et al. Chat-Scene: Bridging 3D Scene and Large Language Models with Object Identifiers. NeurIPS 2024.
>
> [5] Zheng et al. Video-3D LLM: Learning Position-Aware Video Representation for 3D Scene Understanding. CVPR 2025.
>
> [6] Schult et al. Mask3D: Mask Transformer for 3D Semantic Instance Segmentation. ICRA 2023.

---

### Official Review · Reviewer_XwcE · 2025-11-02

**Soundness:** 3
**Presentation:** 3
**Contribution:** 3
**Rating:** 4
**Confidence:** 5

**Summary:**

This work proposes a 3D vision-language model (3D-VLM) called LEO-VL. To reduce the number of tokens representing a scene, it introduces a method named CFG, which back-projects 2D image features into a voxelized 3D space and pools voxel features along the height dimension. The authors also curate existing datasets and generate new samples to construct a 700K 3D-VL instruction-following dataset. Finally, they introduce SceneDPO as an effective post-training objective for 3D-VLMs. Experimental results demonstrate that LEO-VL achieves state-of-the-art performance on several existing benchmarks.

**Strengths:**

1. The motivation to reduce scene tokens is reasonable and addresses an important problem in the field.
2. LEO-VL achieves state-of-the-art results on several benchmarks.
3. The paper also introduces a new 3D-VL dataset, which adds value to the community.

**Weaknesses:**

1. My main concern lies in the fairness of the benchmark comparisons. Although LEO-VL achieves state-of-the-art results on several benchmarks, its VLM backbone and training data differ from those of the compared baselines. Therefore, it is difficult to verify whether the proposed CFG is truly more efficient and effective than previous methods.
2. Regarding the evaluation benchmarks, although it is common practice in the 3D-VL field to use image captioning metrics such as CIDEr and machine translation metrics such as BLEU-4 (B-4) for the ScanQA task, I do not think these are reliable indicators of model performance. These metrics can be easily influenced by the length of the ground-truth answers and synonym variations, which reduces their validity for QA tasks.
For example, while LEO-VL outperforms 3D-LLaVA by +9.2 CIDEr points (101.4 vs. 92.6), it actually performs –3.9 points worse on BLEU-4 (13.2 vs. 17.1). Since both ScanQA and SQA3D are essentially question-answering tasks, the evaluation should rely on accuracy—measuring how many questions are answered correctly. This can be easily computed by having an LLM verify whether the model’s output matches the ground truth. I would like to see results reported in terms of accuracy.
3. Figure 3 is a key experiment intended to demonstrate the effectiveness of the proposed CFG. However, presenting results only on SQA3D is not sufficiently convincing. I recommend including results on additional benchmarks, particularly those reporting accuracy-based metrics, to strengthen the evidence.
4. In Table 6, the performance improvement of SceneDPO over SFT on SQA3D is only 0.5 points, which raises doubts about the effectiveness of SceneDPO.
5. From Lines 227–229, the paper states that the 3D object grounding task is excluded simply because it has a different formulation. However, this explanation is not sufficiently justified, especially since the proposed LEO-VL framework should, in principle, be compatible with object grounding as well.

**Questions:**

/

---

> ### Author Response · Authors · 2025-11-25
> **Rebuttal to Reviewer XwcE (Part 1)**
>
> Thanks for your valuable feedback and constructive comments. We appreciate the positive recognition of our work's strengths, including **good motivation**, **strong experimental results**, and **data-front contribution**. We will address the reviewer's concerns as below.
>
> ---
>
> > 1. My main concern lies in the fairness of the benchmark comparisons. Although LEO-VL achieves state-of-the-art results on several benchmarks, its VLM backbone and training data differ from those of the compared baselines. Therefore, it is difficult to verify whether the proposed CFG is truly more efficient and effective than previous methods.
>
> **`A1:`** Thanks for your feedback! We present two additional results with fairer comparisons regarding the model performance and efficiency, respectively.
>
> **Model performance.** To eliminate the influence of VLM backbone and training data, we directly train Qwen2.5-VL (w/o CFG) on the ScanNet subset of LEO-VL data, comparing to the performance of LEO-VL (ScanNet). The results demonstrate that LEO-VL, equipped with CFG representation, achieves better performance compared to the vanilla VLM backbone when trained with the same data. We also incorporate GPT-Score on ScanQA and SQA3D according to the reviewer's preference.
>
> | ScanNet | ScanQA (CIDEr) | ScanQA (BLEU-4) | ScanQA (GPT-Score) | SQA3D (EM) | SQA3D (EM-R) | SQA3D (GPT-Score) | MSQA (GPT-Score) | Beacon3D (Case) | Beacon3D (Obj.) |
> | ------ | ---- | ---- | ---- | ---- | ---- | ---- | ---- | ---- | ---- |
> | Qwen2.5-VL (w/o CFG) | 96.6 | 16.0 | 44.4 | 54.1 | 57.1 | 58.0 | 56.1 | 55.4 | 17.4 |
> | LEO-VL (w/ CFG) | 101.1 | 16.2 | 47.3 | 58.7 | 61.8 | 62.8 | 58.4 | 58.0 | 17.7 |
>
> **Model efficiency.** We compare the inference efficiency of LEO-VL (w/ CFG) with (1) LLaVA-3D [1], (2) Video-3D LLM [2], and (3) the vanilla VLM backbone Qwen2.5-VL (w/o CFG). Metrics are reported on ScanNet scene0050_00 using the same prompt on an NVIDIA 4090 GPU. The results show that **LEO-VL (w/ CFG) exhibits the highest efficiency, especially in FLOPs and computation time**.
>
> | Inference Efficiency  | FLOPs (T) | Time (s) | Memory (GB) |
> | ------ | ---- | ---- | ---- |
> | LLaVA-3D [1] | 23.3 | 0.240 | 14.961 |
> | Video-3D LLM [2] | 116.7 | 1.375 | 21.689 |
> | Qwen2.5-VL (w/o CFG) | 114.1 | 1.122 | 17.947 |
> | LEO-VL (w/ CFG) | 13.8 | 0.170 | 14.959 |
>
> ---
>
> > 2. Regarding the evaluation benchmarks, although it is common practice in the 3D-VL field to use image captioning metrics such as CIDEr and machine translation metrics such as BLEU-4 (B-4) for the ScanQA task, I do not think these are reliable indicators of model performance. These metrics can be easily influenced by the length of the ground-truth answers and synonym variations, which reduces their validity for QA tasks. For example, while LEO-VL outperforms 3D-LLaVA by +9.2 CIDEr points (101.4 vs. 92.6), it actually performs –3.9 points worse on BLEU-4 (13.2 vs. 17.1). Since both ScanQA and SQA3D are essentially question-answering tasks, the evaluation should rely on accuracy—measuring how many questions are answered correctly. This can be easily computed by having an LLM verify whether the model’s output matches the ground truth. I would like to see results reported in terms of accuracy.
>
> **`A2:`** Good point! We agree with the reviewer on the deficiency of n-gram metrics for evaluating QA performance. The metrics of MSQA and Beacon3D are GPT-Score while ScanQA and SQA3D are not. So we present GPT-Score of ScanQA and SQA3D here, comparing to LLaVA-3D [1] and Video-3D LLM [2]. As shown in the table below, LEO-VL achieves the highest GPT-Score on SQA3D while ranking second on ScanQA. We note that the test data of ScanQA can be imperfect [3], which may undermine the evaluation results.
>
> |  | ScanQA (CIDEr) | ScanQA (BLEU-4) | ScanQA (GPT-Score) | SQA3D (EM) | SQA3D (EM-R) | SQA3D (GPT-Score) |
> | ------ | ---- | ---- | ---- | ---- | ---- | ---- |
> | LLaVA-3D [1] | 91.7 | 14.5 | 43.9 | 55.6 | 57.6 | 56.7 |
> | Video-3D LLM [2] | 100.5 | 16.3 | 50.5 | 57.7 | - | 62.4 |
> | LEO-VL | 100.4 | 15.5 | 47.4 | 60.8 | 63.7 | 64.7 |
>
> ---
>
> > 3. Figure 3 is a key experiment intended to demonstrate the effectiveness of the proposed CFG. However, presenting results only on SQA3D is not sufficiently convincing. I recommend including results on additional benchmarks, particularly those reporting accuracy-based metrics, to strengthen the evidence.
>
> **`A3:`** Good suggestion! We initially focused on SQA3D because it provides the **most complete set of results across competing models**, maximizing the number of models visible in Figure 3. In other words, adding more benchmarks would lead to fewer models in the figure. We understand the need for stronger evidence and will try to reach a better trade-off between **model completeness and metric breadth**.

---

> ### Author Response · Authors · 2025-11-25
> **Rebuttal to Reviewer XwcE (Part 2)**
>
> > 4. In Table 6, the performance improvement of SceneDPO over SFT on SQA3D is only 0.5 points, which raises doubts about the effectiveness of SceneDPO.
>
> **`A4:`** Good question! We clarify that the effectiveness of our approach is primarily demonstrated in two critical aspects:
> - **Improved generalization.** While SFT is known for strong in-domain performance, SceneDPO still outperforms SFT on in-domain SQA3D results. Crucially, SceneDPO shows **significantly superior performance in out-of-domain (OOD) scenarios** (Table 6).
> - **Mitigation of overfitting.** SceneDPO is highly effective in mitigating overfitting (Figure 8). SceneDPO discourages the model from simply memorizing answers and encourages it to properly utilize the visual input, thereby addressing the **common issue of visual ignorance** observed in several recent VLMs [3,4,5,6].
>
> Finally, we conjecture that the relatively small in-domain margin may be due to a **bottleneck in the base model's spatial understanding capabilities**, which hinders SceneDPO from achieving a larger, more distinguishable performance gap over SFT.
>
> ---
>
> > 5. From Lines 227–229, the paper states that the 3D object grounding task is excluded simply because it has a different formulation. However, this explanation is not sufficiently justified, especially since the proposed LEO-VL framework should, in principle, be compatible with object grounding as well.
>
> **`A5:`** Good suggestion! We finetune LEO-VL on ScanRefer and Nr3D datasets, requiring the model to directly output 3D object bboxes in the format $[x_{center}, y_{center}, z_{center}, l_x, l_y, l_z]$, similar to prior works [1,7]. The results confirm LEO-VL's compatibility and we present key takeaways as below.
> - **Competitive grounding performance.** Despite the inherent difficulty of directly predicting 3D bboxes [1,7,8], LEO-VL achieves 60% Acc@IoU25, confirming the effectiveness of our 3D spatial encoding, despite the vertical condensation. However, a relatively weaker Acc@IoU50 suggests a limitation in finer-grained localization, likely due to the compression in representation.
> - **Improvement with object proposals.** Some 3D VLMs [2,9] leverage object proposals (*e.g.*, Mask3D [10]) to simplify the object grounding task as a multi-choice task. We also report results where we retrieve the Mask3D object proposal with the maximum IoU with the model's raw prediction as the final prediction. This approach shows significantly higher accuracies, demonstrating great improvements when aided by object proposals.
>
> | Grounding | ScanRefer (Acc@IoU25) | ScanRefer (Acc@IoU50) | Nr3D (Acc@IoU25) | Nr3D (Acc@IoU50) |
> | ------ | ---- | ---- | ---- | ---- |
> | ReGround3D (w/o proposal) [8] | 53.1 | 41.1 | - | - |
> | LLaVA-3D (w/o proposal) [1] | 50.1 | 42.7 | - | - |
> | Chat-Scene (w/ proposal) [9] | 55.5 | 50.2 | - | - |
> | Video-3D LLM (w/ proposal) [2] | 57.9 | 51.2 | - | - |
> | LEO-VL (w/o proposal) | 60.0 | 20.6 | 48.3 | 15.1 |
> | LEO-VL (w/ proposal) | 83.7 | 75.2 | 73.0 | 63.3 |
>
> ---
>
> **Thank you once again for the time dedicated to reviewing our work. We look forward to any further discussion or guidance.**
>
>
> [1] Zhu et al. LLaVA-3D: A Simple yet Effective Pathway to Empowering LMMs with 3D Capabilities. ICCV 2025.
>
> [2] Zheng et al. Video-3D LLM: Learning Position-Aware Video Representation for 3D Scene Understanding. CVPR 2025.
>
> [3] Huang et al. Unveiling the Mist over 3D Vision-Language Understanding: Object-centric Evaluation with Chain-of-Analysis. CVPR 2025.
>
> [4] Wang et al. mDPO: Conditional Preference Optimization for Multimodal Large Language Models. EMNLP 2024.
>
> [5] Xie et al. V-DPO: Mitigating Hallucination in Large Vision Language Models via Vision-Guided Direct Preference Optimization. EMNLP 2024 Findings.
>
> [6] Brown et al. Benchmark Designers Should "Train on the Test Set" to Expose Exploitable Non-Visual Shortcuts. arXiv, 2025.
>
> [7] Chen et al. LL3DA: Visual Interactive Instruction Tuning for Omni-3D Understanding, Reasoning, and Planning. CVPR 2024.
>
> [8] Zhu et al. ScanReason: Empowering 3D Visual Grounding with Reasoning Capabilities. ECCV 2024.
>
> [9] Huang et al. Chat-Scene: Bridging 3D Scene and Large Language Models with Object Identifiers. NeurIPS 2024.
>
> [10] Schult et al. Mask3D: Mask Transformer for 3D Semantic Instance Segmentation. ICRA 2023.

---

### Author Response · Authors · 2025-12-01
**Summary to AC/SAC**

Dear AC/SAC,

We thank the reviewers for their time and engagement. Due to the closure of the author-reviewer discussion phase, we summarize how our rebuttal addresses the initial concerns. We frame this summary per reviewer and use P1/P2/P3... to denote the concerns.

- **R1 (XwcE): rating 4, not responded.** Among R1's five points, P1 and P5 also appear in R3/R4's reviews, and both R3 and R4 agreed that our rebuttal addressed them. The remaining P2/P3/P4 involve evaluation metrics and analysis, and we have provided additional results and detailed clarifications in our rebuttal.
- **R2 (DVCP): rating 6, not responded.** Of the three points raised, P1/P2 overlap with R3's concerns, and R3 confirmed our rebuttal addressed them. P3 is an acknowledged limitation already discussed in our manuscript, which we believe does not undermine our contributions and can be left to future work.
- **R3 (YEFT): rating 4 $\rightarrow$ 8, responded.** The reviewer agreed that our rebuttal fully resolved all three concerns, including citation (P1), representation capacity and compatibility with the grounding task (P2), and implementation details (P3).
- **R4 (MyZJ): rating 2 $\rightarrow$ 4, responded but not converged.** We appreciate the reviewer's engagement. While R4 acknowledged that our rebuttal addressed the three initial concerns, he/she raised a new concern (*"refining previous methods..."*) as the sole impediment to a higher score. We elaborated on our contributions and insights regarding this remaining concern. While we believe our elaboration could address this point, we could not receive further confirmation from R4 due to the discussion closure.

We have revised the manuscript to include updated related work and additional experimental results. For concerns that were shared across reviewers (R1/R2 concerns overlapping with R3/R4's resolved issues), we respectfully invite the AC/SAC to consider the positive acknowledgment from R3/R4. For unresolved points, we hope the AC/SAC will consider our response and evaluate the merits of our contributions.

---

### Meta-Review · Area_Chair_Sj4A · 2026-01-06

**Summary:**

By carefully reviewing the reviewers’ comments and the authors’ responses, I agree with the major concerns raised during the review process. In particular, (1) the evaluation relies on an unfair comparison that uses a stronger baseline trained with a larger dataset, and (2) the proposed method is only evaluated on indoor scenes, without validation on more diverse and challenging outdoor scenarios. These issues are not fully addressed in the rebuttal and substantially limit the strength of the empirical claims. Given that ICLR is a top-tier venue, I therefore recommend rejection.

**Reviewer Concerns:**

The reviewer XwcE raises concerns regarding the fairness of the experimental evaluation. After carefully examining the rebuttal, I believe that these core concerns are not fully addressed by the authors. In particular, in response to W1 raised by Reviewer XwcE, the authors do not provide sufficiently direct or solid experimental evidence, e.g., comparing the proposed methods with counterparts using the same baseline and the same training data. This substantially weakens the persuasiveness of the paper.

**Reviewer Scores:**

The initial review scores are 4/2/6/4, with three reviews being negative and one positive. After carefully examining the rebuttal, I find that one major concern has not been convincingly addressed by the authors. So, I believe that the reviewers would likely maintain their overall negative stance toward this submission.

---

### Decision · Program_Chairs · 2026-01-26

Reject